# URDF-Anything: Constructing Articulated Objects with 3D Multimodal Language Model

**Zhe Li**[1,*], **Xiang Bai**[1,*], **Jieyu Zhang**[2], **Zhuangzhe Wu**[1], **Che Xu**[1],
**Ying Li**[1], **Chengkai Hou**[1], **Shanghang Zhang**[1,†]

[1]State Key Laboratory of Multimedia Information Processing, School of Computer
Science, Peking University, [2]University of Washington

[*]Equal contribution and co-first authors.     [†]Corresponding author.

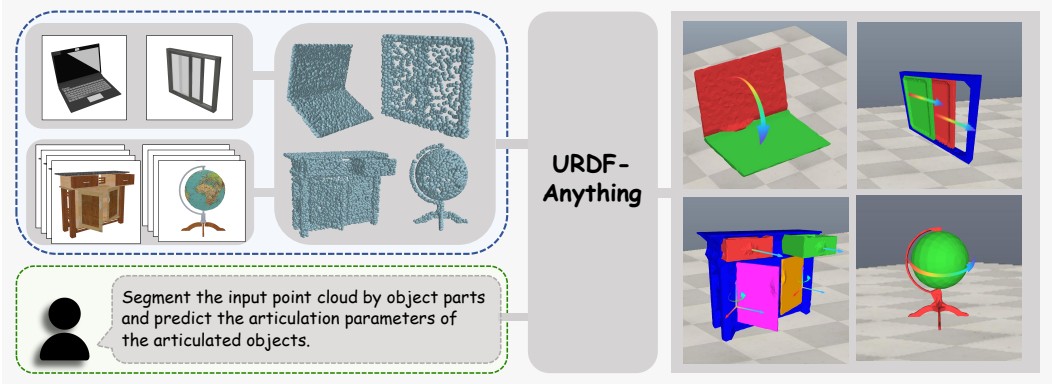

Figure 1: URDF-Anything: Generating Functional URDF Digital Twins from Visual Observations(single or multi-view images). Our framework, utilizing a 3D Multimodal Large Language Model and guided by instructions (e.g., "Segment parts and predict parameters"), processes the point cloud to jointly infer geometric part segmentation and kinematic structure. The output is a segmented 3D model with defined joints (represented here by different part colors), forming a functional URDF digital twin directly usable in physics simulators.

## Abstract

Constructing accurate digital twins of articulated objects is essential for robotic simulation training and embodied AI world model building, yet historically requires painstaking manual modeling or multi-stage pipelines. In this work, we propose **URDF-Anything**, an end-to-end automatic reconstruction framework based on a 3D multimodal large language model (MLLM). URDF-Anything utilizes an autoregressive prediction framework based on point-cloud and text multimodal input to jointly optimize geometric segmentation and kinematic parameter prediction. It implements a specialized $[SEG]$ token mechanism that interacts directly with point cloud features, enabling fine-grained part-level segmentation while maintaining consistency with the kinematic parameter predictions. Experiments on both simulated and real-world datasets demonstrate that our method significantly outperforms existing approaches regarding geometric segmentation (mIoU 17% improvement), kinematic parameter prediction (average error reduction of 29%), and physical executability (surpassing baselines by 50%). Notably, our method exhibits excellent generalization ability, performing well even on objects outside the training set. This work provides an efficient solution for constructing digital twins for robotic simulation, significantly enhancing the sim-to-real transfer capability.

---

[1]Project page: `https://lzvsdy.github.io/URDF-Anything/`

# 1 Introduction

Constructing high-fidelity digital twins is essential for accurately simulating real-world physical dynamics and complex interactions across numerous fields. These include core applications in robotic simulation and training[1, 2] as well as expanding domains like autonomous driving[3, 4, 5] and interactive virtual/augmented reality environments[6, 7, 8]. However, populating these interactive virtual environments requires precise digital representations of their components, especially articulated objects (doors, drawers, scissors) with their complex internal structures and diverse degrees of freedom, which typically demands painstaking manual modeling, and extensive development time. Automating the reconstruction of these objects into fully functional, high-fidelity digital twins (for example, in URDF format) not only alleviates this laborious effort but also offers an efficient pathway to enrich virtual worlds and enable robust simulation-to-real transfer in robotic training.

However, automatically reconstructing articulated objects from visual observations presents significant challenges. Unlike rigid objects, articulated objects consist of multiple parts connected by joints, introducing complexities related to inferring both the geometry of individual links and the kinematic parameters (type, origin, axis, limits) governing their motion. Prior efforts reconstruct articulated objects by composing multiple models into a sophisticated pipeline and either rely on a given mesh assets library or involve a separate part segmentation stage [9, 10].

In this paper, we explore an end-to-end approach for generating functional URDF models directly from visual input; we propose **URDF-Anything**, which leverages 3D Multimodal Large Language Models (MLLMs) to jointly interpret object geometry and semantic attributes, infer kinematic structure, and automatically produce high-fidelity URDF descriptions (Figure 2 illustrates the overall pipeline). A 3D MLLM is uniquely suited for this task, as it can natively handle multimodal input (visual features and text instructions), encodes powerful priors on 3D shapes from large-scale pretraining, and directly understands spatial relationships to output precise coordinates. These capabilities make it ideal for predicting detailed kinematic parameters and generalizing to unseen objects. In addition, we leverage a dynamic $[SEG]$ token mechanism [11] within the MLLM's autoregressive generation, which enables simultaneous prediction of the symbolic articulated structure (link names, joint parameters) and emission of explicit segmentation signals that guide geometric segmentation of object parts from point-cloud features via cross-attention. This tight coupling between symbolic output and geometric segmentation, combined with end-to-end training, ensures full consistency between predicted kinematics and reconstructed geometry, and enables an end-to-end approach for reconstructing articulated objects.

Experimental evaluation on the PartNet-Mobility dataset [12], including both in-distribution and challenging out-of-distribution objects, demonstrates the superior performance of URDF-Anything compared to existing methods. Quantitatively, for part-level geometric segmentation,we achieve higher mIoU on OOD instances (0.62) compared to the best baseline (0.51), and a much higher Count Accuracy (0.97), surpassing the best baseline (0.84) by over 15 points. In joint parameter prediction, our method consistently achieves significantly lower errors across all object categories compared to baselines, outperforming them by a considerable margin, particularly on OOD objects. Crucially, the physical executability rate of our generated URDF models is substantially higher than baselines(50% improvement), enabling more robust simulation for unseen objects. These comprehensive results highlight the effectiveness and strong generalization capability of our end-to-end MLLM framework for automated articulated object reconstruction.

In summary, our main contributions are as follows:

- Proposing the first end-to-end 3D MLLM framework for articulated object reconstruction, championing a new paradigm from complete 3D input to joint prediction output.
- Achieving deep coupling and joint prediction of kinematic parameters and geometric segmentation through an innovative application of the [SEG] token.
- Demonstrating the superiority of this new paradigm through extensive experiments.

# 2 Related Work

**LLMs for 3D Tasks.** Recent years have seen significant progress in applying Large Multimodal Models (LMMs) to 3D understanding and interaction tasks [13, 14, 15, 16, 17, 18, 19, 20, 21]. These models leverage multimodal inputs (such as point clouds or images, combined with language) for 3D

spatial reasoning, perception, and structured output generation. Relevant advancements include the development of 3D MLLM backbones like ShapeLLM [22], capable of handling point cloud-language interaction and demonstrating strong 3D understanding. Other works have explored language-guided 3D segmentation [23], open-world 3D understanding [24], and tasks requiring precise 3D spatial understanding from visual input [25]. While these models demonstrate powerful capabilities in general 3D perception and generating text or segmentation masks, they do not typically address the complex, coupled task of jointly inferring detailed link geometry and precise kinematic parameters for articulated objects. Our work builds upon these advancements in 3D LMMs and their capabilities for spatial reasoning and multimodal generation, specifically applying them to the challenging task of articulated object reconstruction.

**Articulated object modeling.** The automated modeling of articulated objects for robotic manipulation is an active field of research, with several distinct methodological approaches. Physics-based interactive methods [26, 27, 28] utilize interaction in simulation or the real world to refine or build models, demonstrating high accuracy with sufficient interaction but often requiring initial models, struggling with robust passive reconstruction (without interaction), or limited adaptation to novel geometries without extensive interaction data. Automation methods [10, 9] leveraging visual-language models automate digital twin creation, using techniques such as VLM-to-code generation, iterative refinement, mesh retrieval [10], or abstracting parts as OBBs with LLM parameter prediction [9]. However, these methods can be limited by constraints such as reliance on asset databases, brittleness in iterative processes, or loss of geometric detail and potential parameter inaccuracy from using simplified representations like OBBs. [29] is fundamentally limited by its reliance on a hard-coded system to assign kinematic parameters and retrieve meshes based on the network's discrete part classification output, a method which compromises final geometric and kinematic fidelity. [30] is a specialized, small-scale feed-forward model with severe input and output restrictions. Other works [31, 32, 33, 34, 35] explore novel representations like 3D Gaussians for reconstruction or focus on detection of specific features [36], addressing sub-problems rather than the complete, end-to-end geometry and kinematics inference pipeline needed for a functional URDF. Our work contributes to the field of automated articulated object modeling by proposing a novel end-to-end framework that leverages the capabilities of 3D Large Multimodal Models for robust reconstruction from visual input. While recent works have explored VLMs or 2D-based LLMs for URDF generation [10, 9], none have leveraged raw 3D point clouds as the primary input to an MLLM for end-to-end URDF synthesis—a key enabler of geometric precision in our framework.

# 3 Method

URDF-Anything is an end-to-end framework for reconstructing articulated objects from visual observations into URDF-formatted digital twins. The pipeline consists of three main stages: (1) Input Representation, where we generate dense 3D point clouds from single or multi-view RGB images, in Sec3.2 ; (2) Multimodal Articulation Parsing, where a 3D Multimodal Large Language Model (MLLM) jointly predicts part segmentation and kinematic parameters, in Sec3.3 ; and (3) Mesh Conversion, where segmented point clouds are converted into meshes for simulation, in Sec3.5. The key innovation lies in the integration of geometric and semantic features through a dynamic $[SEG]$ token mechanism, enabling precise part-level segmentation and joint parameter prediction in a unified framework.

## 3.1 Task Definition

To ensure compatibility with standard 3D simulators (e.g., MuJoCo, PyBullet), we represent articulated objects as URDF (Unified Robot Description Format) models. A URDF structure defines an object as a hierarchical tree composed of:

**Links**: Rigid components representing object parts (e.g., a cabinet's door, drawer, or base). Each link contains geometric (mesh) and inertial properties.

**Joints**: Connections between links that specify kinematic constraints. Each joint includes:

- Type: Prismatic, revolute, continuous, floating, planar or fixed.
- Parent/Child: Links connected by the joint.
- Origin : 3D position $(x, y, z) \in \mathbb{R}^3$ and orientation $(r, p, y) \in \mathbb{R}^3$ relative to the parent link.

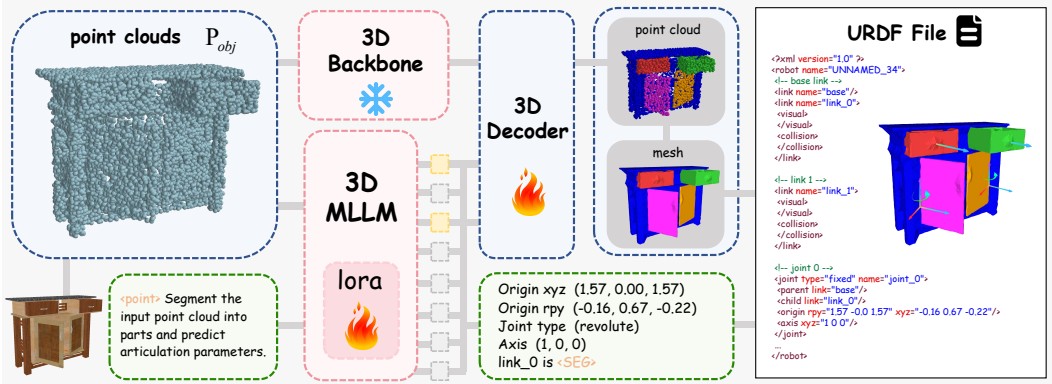

Figure 2: Overview of the URDF-Anything Framework. The pipeline takes a 3D point cloud (from image) and a structured language instruction as input. The 3D MLLM(fine-tuned with LoRA) autoregressively generates symbolic output (kinematic parameters) and $[SEG]$ tokens. The embeddings corresponding to the generated $[SEG]$ tokens then interact with the point cloud features via a 3D Decoder to perform fine-grained geometric segmentation of the point cloud into individual links. Finally, the jointly predicted kinematic parameters and the segmented geometry are integrated into a functional URDF file, resulting in a complete articulated 3D model ready for physics simulation.

- Axis: Normalized 3D vector defining the motion direction (e.g., $[0, 1, 0]$ for vertical sliding).
- Limit: Defines the range of motion for prismatic, revolute and planar joints.

Our objective is to automatically infer these URDF components—specifically, the mesh for each link, the type, origin, parent, child, and axis for each joint—from visual input of the articulated object.

### 3.2 Input Representation

The initial step of our pipeline is to transform the raw visual observation of an articulated object into a 3D representation suitable for structural parsing. Real-world observations can vary significantly, sometimes providing multiple viewpoints of the object and other times offering only a single view. Accordingly, our point cloud acquisition strategy adapts to the input modality:

**Multi-view Input:** When multiple RGB images from different viewpoints are available , we leverage DUSt3R[37] to generate a dense 3D point cloud $P_{obj}$. DUSt3R excels at establishing dense 2D-to-3D correspondences and reconstructing accurate 3D geometry from multiple views.

**Single-view Input:** For the more challenging single-view scenario, we utilize a generative approach LGM[38]. We first employ a pre-trained diffusion model to synthesize consistent multi-view images. Subsequently, we reconstruct the 3D geometry from these synthesized views, which can be represented as a 3D Gaussian Splatting model or converted into a point cloud $P_{obj}$.

Regardless of the input type (single or multi-view), the output of this stage is a dense point cloud $P_{obj} \in \mathbb{R}^{N \times 6}$ representing the entire articulated object. It is crucial to note that this initial point cloud is a monolithic representation; it captures the overall geometry but does not inherently differentiate or segment the object into its individual links or sub-components. This whole-object point cloud $P_{obj}$ serves as the primary geometric input for the subsequent multimodal articulation parsing stage. More details about this section are reported in Appendix A.2.

### 3.3 Articulation Parsing with 3D MLLM

We adopt ShapeLLM [22], a recent 3D MLLM, as our backbone, which combines a point cloud encoder[39] with a large language model[40]. ShapeLLM has demonstrated the ability to perform 3D visual grounding and output structured 3D information, such as 3D bounding-box coordinates. This pre-existing capability is particularly beneficial for our task of predicting joint parameters, which inherently involve 3D locations (origin, axis) and orientations (origin). This unified architecture offers two advantages: (1) Open-world generalization: The 3D MLLM's geometric understanding capability, enhanced by large-scale 3D-language pretraining, enables robust reasoning about both seen and

unseen object categories. (2) Structured output capability: The MLLM inherently supports generating JSON-formatted outputs of arbitrary length and complexity, enabling flexible representation of joint hierarchies and part relationships.

**Input:** The input point cloud $P_{obj}$ is encoded by a 3D encoder to extract dense geometric features $F_{pc} \in \mathbb{R}^{M \times d_{pc}}$, where $M$ is the number of points (potentially downsampled) and $d_{pc}$ is the feature dimension. Simultaneously, natural language instructions are provided to guide the parsing process. We employ a structured instruction template that explicitly incorporates the geometric features and prompts the model for specific URDF information. The textual part of the instruction, $X_{txt}$, is processed by the LLM's standard word embedding layer to yield text embeddings $F_{txt} \in \mathbb{R}^{L \times d_{txt}}$, where $L$ is the number of tokens and $d_{txt}$ is the embedding dimension. These multimodal features ($F_{pc}$ and $F_{txt}$) are integrated and processed by the LLM's layers. This core multimodal processing and output generation can be formally represented as:

$$Y_{output} = \text{MLLM}(F_{pc}, F_{txt})$$

**Output:** While 3D MLLMs excel at processing multimodal inputs and generating structured textual outputs, they are not inherently equipped to generate dense, per-point predictions required for geometric segmentation. Our task necessitates not only predicting the symbolic URDF structure and parameters but also precisely segmenting the corresponding point cloud geometry for each link. Inspired by LISA[11], we extend the vocabulary with a special token $[SEG]$ to achieve this joint objective. The MLLM is trained to autoregressively generate a structured JSON output sequence that contains the predicted joint parameters alongside descriptions of the links. Crucially, each link description in the output is associated with a $[SEG]$ token (e.g., "link_0": "base_cabinet$[SEG]$", "link_1": "drawer$[SEG]$"). This means the MLLM simultaneously predicts the symbolic representation of the articulated structure and places markers ($[SEG]$) indicating the geometry corresponding to each part needs to be identified. More details are reported in Appendix A.3.

### 3.4 Geometric Segmentation from Special-Token Mechanism

Geometric segmentation is performed for each object part indicated by a $[SEG]$ token in the MLLM's output sequence $Y_{output}$. For each generated $[SEG]$ token, we leverage its final hidden state $h_{seg}$, combined with the preceding part category token's state $h_{category}$ to form a fused token representation $h_{combined} = [h_{category}; h_{seg}]$. This representation is then used as the query

$$H_{query} = \text{MLP}_{query}(h_{combined})$$

in a cross-attention mechanism. Another point feature $S_{pc}$ is generated from 3D backbone $S_{pc} = F_{enc}(P_{obj})$. Then a mechanism attends over the projected point cloud features $F'_{pc} = \text{MLP}_{pc}(S_{pc})$, effectively computing per-point scores

$$y_{mask} = \text{CrossAttn}(Q = H_{query}, K = F'_{pc}, V = F'_{pc})$$

indicating the likelihood of each point belonging to the part. These scores $y_{mask}$ are converted into a binary segmentation mask for the corresponding link via a sigmoid and threshold. This process is repeated for every $[SEG]$ token, yielding masks for all predicted parts.

### 3.5 Mesh Conversion and URDF File Generation

The final stage of our pipeline consolidates the reconstructed geometry and kinematics into a standard URDF XML file, ready for direct use in physics simulators. Segmented point clouds for each link, obtained from the geometric segmentation process (Section 3.4), are converted into 3D mesh representations (e.g., OBJ format) using point-to-mesh conversion method[41, 42]. Simultaneously, the MLLM's structured JSON output provides the complete kinematic structure, including joint types, parent/child relationships, origins, and axis. We parse this JSON and assemble the final URDF XML file, where each link references its generated mesh and joint elements are populated with the predicted parameters. The resulting URDF model can be directly imported into standard physics simulators (e.g., MuJoCo [43], Sapiens [44]). A complete example of a generated URDF file for a sample object instance is provided in Figure8.

### 3.6 Model Training

The URDF-Anything model is trained end-to-end to jointly generate the structured URDF parameters and predict accurate part-level segmentation masks from the point cloud. The overall training objective $L$ is a weighted sum of the language modeling loss $L_{text}$ and the segmentation loss $L_{seg}$:

$$L = \lambda_{text}L_{text} + \lambda_{seg} \sum_{i=1}^{N} L_{i,seg}$$

where $\lambda_{lm}$ and $\lambda_{seg}$ are hyperparameters balancing the contribution of each loss, and $N$ is the num of object parts. The segmentation loss uses a combination of binary cross-entropy (BCE) loss and Dice loss (DICE). For a given part associated with a $[SEG]$ token, let $M_{gt} \in \{0,1\}^M$ be the ground truth binary mask for that part. The segmentation loss for this part is:

$$L_{seg} = \lambda_{bce}\text{BCE}(\hat{M}, M_{gt}) + \lambda_{dice}\text{DICE}(\hat{M}, M_{gt})$$

where $\lambda_{bce}$ and $\lambda_{dice}$ are hyperparameters weighting the BCE and DICE components.

## 4 Experiments

**Implementation Details:** We employ ShapeLLM [22] as our 3D MLLM backbone, with ShapeLLM-7B-general-v1.0 checkpoint as the default settings. For the 3D backbone, We use Uni3D [45] to extract dense geometric features. We adopt one NVIDIA 80G A800 GPU for training. We employ LoRA [46] for efficient fine-tuning, with the rank of LoRA set to 8 by default. We use AdamW optimizer [47] with the learning rate and weight decay set to 0.0003 and 0, respectively. We use the cosine learning rate scheduler, with the warm-up iteration ratio set to 0.03. The batch size per device is set to 2, and the gradient accumulation step is set to 10. Our model was fine-tuned in 2.5 hours on a single NVIDIA A800 (80GB) GPU.

**Dataset:** We train and evaluate our framework on the PartNet-Mobility dataset [12], a large collection of 3D articulated objects with URDF annotations. The dataset is partitioned into In-Distribution (ID) and Out-of-Distribution (OOD) subsets based solely on object categories. Since our method utilizes image input, we generated visual data by rendering multi-view and single-view RGB images from the dataset's 3D models within a simulation environment. Ground truth kinematic and geometric information from the original URDF files was processed and reorganized into a compact JSON format matching our model's output structure. The dataset is divided into standard training and testing sets. More details are reported in Appendix A.1.

**Baselines:** We compare against three prior methods: (1)**Articulate-Anything,** an actor-critic system for iterative refinement via a mesh retrieval mechanism to generate code that can be compiled into simulators. (2) **Real2Code:** an approach that represents object parts with oriented bounding boxes, and uses a fine-tuned LLM to predict joint parameters as code. (3)**URDFormer:** a pipeline that constructs simulation scenes complete with articulated structure directly from real-world images.

**Evaluation Metrics:** We evaluate the performance of our URDF-Anything framework on three key aspects: part-level geometric segmentation, kinematic parameter prediction accuracy, and the physical executability of the final reconstructed URDF.

- **Part Segmentation Accuracy:** For evaluating how accurately our method segments the point cloud into individual links, we use mIoU. A higher mIoU indicates better alignment between predicted and ground truth part geometry. We use Count Acc to measure the percentage of test samples where the number of predicted articulated parts (i.e., links with non-fixed joints) exactly matches the ground-truth count.

- **Joint Parameter Prediction Accuracy:** Following Articulate-Anything[10], we evaluate the correctness of several critical components for each joint prediction compared to the ground truth URDF: (1)**Joint Type Error:** A binary metric indicating if the predicted joint type (e.g., revolute, prismatic) matches the ground truth. (2)**Joint Axis Error:** Quantifies the angular difference between the predicted and ground truth joint axes, normalized to be within $[0, \pi]$. (3)**Joint Origin Error:** Measures the positional error of the joint origin.

- **Physical Executability:** Beyond static accuracy metrics, the ultimate test of a reconstructed URDF is its functionality in a physics simulation. We evaluate the physical executability by

Table 1: **Quantitative Results for Part-Level Link Segmentation.** Comparison of our method ([SEG](Ours)) against baselines (Uni3D w/o text, Uni3D w/ text) on In-Distribution (ID) and Out-of-Distribution (OOD) object instances. Performance is measured by mIoU (segment accuracy) and Count Accuracy (Count Acc), both where higher is better (↑).

| Models | mIoU ↑ | | | Count Acc ↑ | | |
|---|---|---|---|---|---|---|
| | ALL | ID | OOD | ALL | ID | OOD |
| Uni3D w/o text | $0.36 \pm 0.01$ | $0.50 \pm 0.02$ | $0.33 \pm 0.01$ | $0.73 \pm 0.02$ | $0.83 \pm 0.03$ | $0.70 \pm 0.01$ |
| Uni3D w/ text | $0.54 \pm 0.02$ | $0.64 \pm 0.01$ | $0.51 \pm 0.02$ | $0.84 \pm 0.02$ | $0.91 \pm 0.02$ | $0.82 \pm 0.02$ |
| **URDF-Anything ([SEG])** | **0.63** (16.7%↑) | **0.69** $\pm 0.01$ | **0.62** $\pm 0.01$ | **0.97** (15.4%↑) | **0.99** $\pm 0.02$ | **0.96** $\pm 0.02$ |

loading the generated URDF models into a standard 3D simulator (e.g., Sapiens [44]). We then programmatically or manually attempt to actuate the joints. The physical executability metric is defined as the percentage of generated URDFs that can be loaded and actuated correctly in the simulator without exhibiting non-physical behavior (e.g., parts flying off, joints freezing, unexpected rotations/translations). This metric provides a crucial validation of the end-to-end reconstruction quality and its suitability for sim-to-real applications.

## 4.1 Link Segmentation Results

Table 1 presents the quantitative results for part-level geometric segmentation. This evaluation assesses how accurately each method can delineate the individual links of an articulated object within its point cloud representation. We compare our proposed method against two baselines that represent different configurations of utilizing Uni3D features: (1) **Uni3D w/o text**, a supervised closed-set model that adds an MLP head to classify points into predefined part categories using only geometric features; and (2) **Uni3D w/ text**, which follows Uni3D's original text-guided paradigm by aligning point features with a fixed list of part-name prompts (e.g., drawer", base") via a feature propagation layer. Both baselines lack the dynamic, context-aware coupling between segmentation and kinematic structure enabled by our [SEG] token mechanism.

The poor performance of **Uni3D w/o text** demonstrates that relying solely on geometric features is insufficient, highlighting the necessity of semantic guidance. Its weakness stems from a paradigm mismatch—it discards the cross-modal semantic alignment Uni3D was pre-trained for. Meanwhile, our method significantly outperforms **Uni3D w/ text**, showing that our [SEG] token mechanism provides a superior guidance strategy: unlike static text prompting, it dynamically and end-to-end couples segmentation with the autoregressive generation of kinematic structure, yielding more accurate and context-aware results.

Our method achieves state-of-the-art performance across all metrics and datasets. Quantitatively, URDF-Anything ([$SEG$]) significantly outperforms the baseline on average, achieving 0.63 mIoU (a **16.7%** improvement) and 0.97 Count Accuracy (a **15.4%** improvement). This strong performance holds for both ID and OOD data, particularly on challenging OOD instances, demonstrating superior generalization capability compared to baselines. The significant improvement in mIoU on OOD data demonstrates superior generalization capability in segmenting novel object geometries and structures not seen during training.

Table 2: **Quantitative Comparison of Joint Parameter Prediction Accuracy.** This table presents the average prediction errors for joint Type (fraction of incorrect), Axis (radians), and Origin (meters) across different methods. Results are shown for All, In-Distribution (ID), and Out-of-Distribution (OOD) object classes. Lower values indicate better performance (↓). "Oracle" denotes baselines evaluated with ground-truth part segmentation to isolate kinematic prediction performance, following the protocol in [10].

| Method | All Classes | | | ID Classes | | | OOD Classes | | |
|---|---|---|---|---|---|---|---|---|---|
| | Type ↓ | Axis ↓ | Origin ↓ | Type ↓ | Axis ↓ | Origin ↓ | Type ↓ | Axis ↓ | Origin ↓ |
| Real2Code Oracle | $0.537 \pm 0.014$ | $1.006 \pm 0.723$ | $0.294 \pm 0.417$ | $0.410 \pm 0.029$ | $1.164 \pm 0.671$ | $0.344 \pm 0.479$ | $0.576 \pm 0.016$ | $0.937 \pm 0.734$ | $0.272 \pm 0.386$ |
| URDFormer Oracle | $0.556 \pm 0.025$ | $0.374 \pm 0.666$ | $0.581 \pm 0.355$ | $0.418 \pm 0.036$ | $0.208 \pm 0.532$ | $0.609 \pm 0.357$ | $0.679 \pm 0.032$ | $0.643 \pm 0.766$ | $0.513 \pm 0.340$ |
| Articulate-Anything | $0.025 \pm 0.005$ | $0.145 \pm 0.450$ | $0.207 \pm 0.392$ | $0.018 \pm 0.004$ | $0.143 \pm 0.198$ | $0.195 \pm 0.237$ | $0.026 \pm 0.005$ | $0.145 \pm 0.480$ | $0.208 \pm 0.411$ |
| **URDF-Anything (Ours)** | **0.008** $\pm 0.001$ | **0.132** $\pm 0.048$ | **0.164** $\pm 0.026$ | **0.007** $\pm 0.001$ | **0.121** $\pm 0.039$ | **0.130** $\pm 0.022$ | **0.009** $\pm 0.001$ | **0.136** $\pm 0.050$ | **0.173** $\pm 0.027$ |

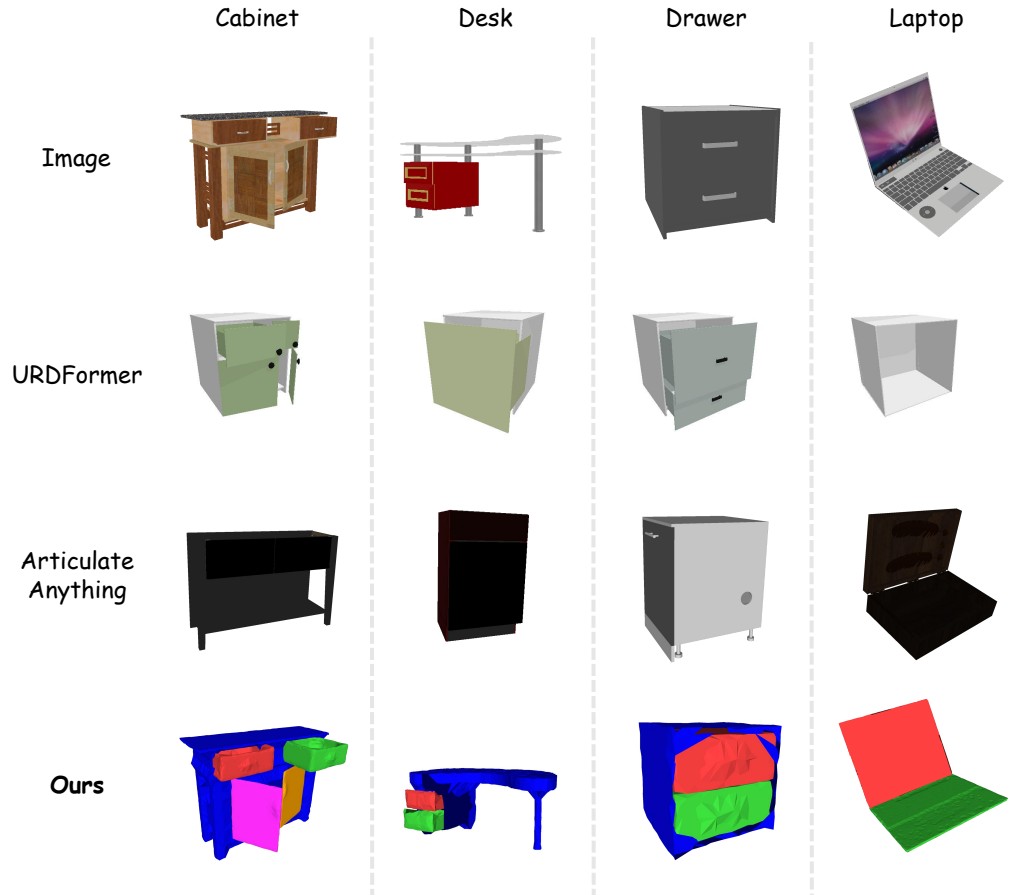

Figure 3: **Qualitative Comparison of Articulated Object Reconstruction Results.** The top row displays the input image for various articulated object instances (each column represents a different object). We can find that baseline methods frequently struggle in predicting incorrect object types, generating distorted geometry, or exhibiting significant errors in link placement, leading to misaligned or incorrect structures.

## 4.2 Joint Parameter Prediction Results

Table 2 presents the quantitative results for joint parameter prediction accuracy. We compare our proposed framework, **URDF-Anything (Ours)**, against several prior methods. As shown in Table 2, URDF-Anything demonstrates superior performance compared to baseline methods across all evaluated categories. Specifically, our method achieves notably lower Type, Axis, and Origin errors when evaluated on All Classes, ID Classes, and particularly on OOD Classes. This indicates a significant improvement in accurately predicting kinematic parameters for articulated objects, including robust performance on object categories not seen during training (OOD instances).

This strong performance on OOD articulated objects highlights the effectiveness of our MLLM-based end-to-end approach. The MLLM's inherent capabilities in understanding complex structures and generalizing from large-scale pretraining, combined with our method's joint reasoning of geometry and kinematics, enables more accurate parameter prediction in challenging, unseen scenarios compared to prior methods that may struggle with complexity or lack strong generalization priors.

## 4.3 Physical Executability

Beyond evaluating static link segmentation and joint parameter accuracy, a critical validation of our end-to-end reconstruction framework is the physical executability of the generated URDF models in a simulation environment. As defined in Section 4, this metric assesses the percentage of generated

Table 3: **Physical Executability Rate (% ↑) across methods on ID and OOD subsets.**

| Method | ALL | ID Classes | OOD Classes |
|---|---|---|---|
| URDFormer Oracle | 24% | 34% | 15% |
| Real2Code Oracle | 41% | 49% | 23% |
| Articulate-Anything | 52% | 61% | 44% |
| **URDF-Anything(Ours)** | **78%**(50.0%↑) | **86%** | **71%** |

Table 4: **Ablation Study on Input Modality for Joint Parameter Prediction.**

| Method Variant | Input Modality | Type ↓ | Axis ↓ | Origin ↓ |
|---|---|---|---|---|
| OBB | Text | $0.42_{\pm0.18}$ | $0.70_{\pm0.26}$ | $0.47_{\pm0.22}$ |
| Point Cloud only | Point Cloud | $0.34_{\pm0.11}$ | $0.29_{\pm0.18}$ | $0.26_{\pm0.15}$ |
| Qwen2.5-VL-7B | Image + Text | $0.57_{\pm0.14}$ | $0.85_{\pm0.31}$ | $0.23_{\pm0.16}$ |
| Qwen2.5-VL-7B + ft | Image + Text | $0.38_{\pm0.09}$ | $0.81_{\pm0.24}$ | $0.18_{\pm0.10}$ |
| **Point Cloud + Text (Ours)** | Point Cloud + Text | $\mathbf{0.008}_{\pm\mathbf{0.001}}$ | $\mathbf{0.132}_{\pm\mathbf{0.048}}$ | $\mathbf{0.164}_{\pm\mathbf{0.026}}$ |

URDFs that can be successfully loaded and actuated in a physics simulator without errors or non-physical behavior.

As shown in Table 3, URDF-Anything achieves a high physical executability rate, significantly surpassing baseline methods, particularly for OOD objects. This demonstrates the superior overall pipeline robustness of our approach. Compared to prior methods like Real2Code [9], which rely on complex, sequential pipelines where errors in one step can cascade and require manual intervention for refinement, or Articulate-Anything [10], which may depend on iterative refinement for parameter estimation, our framework utilizes a unified, end-to-end MLLM that jointly reasons about geometry and kinematics. This direct, integrated approach minimizes error propagation, allowing the model to leverage rich multimodal context for robust prediction of a consistent geometric and kinematic structure in a single pass. Figure 3 and Figure7 provides qualitative results illustrating the visual quality of our reconstructed articulated objects, showcasing both the generated link segmentation and the resulting mesh geometry for various challenging instances.

## 4.4 Ablation Study

**Impact of the geometric representation Modality.** We conducted an ablation study on input modality, with results in Table 4. First, we found that even powerful, fine-tuned image MLLMs (e.g., 'Qwen2.5-VL-7B + ft') struggle to infer precise 3D kinematic parameters from 2D images, yielding significantly higher errors than any 3D-based approach. This highlights the necessity of explicit 3D geometry for this task.

Next, we evaluated different 3D representations. We observed that simplified geometry, such as Oriented Bounding Boxes ('OBB'), is insufficient due to the loss of crucial geometric detail. While using a detailed point cloud alone ('Point Cloud only') improves performance, our full method ('Point Cloud + Text'), which integrates language guidance into MLLM, achieves the best results by a significant margin. This highlights the necessity of language and MLLM. Collectively, this study validates our core design choice: achieving high-fidelity reconstruction requires the combination of detailed 3D geometric input and effective language guidance within a 3D MLLM framework.

**Importance of Joint Geometric and Kinematic Prediction.** A core hypothesis of our work is that jointly predicting an object's geometry and kinematics is superior to tackling these tasks independently. We posit that structured reasoning about kinematics provides crucial context for geometric understanding, and similarly, geometric features must ground the kinematic predictions. To validate this, we compare our full, jointly-trained model against two decoupled variants:

- **Kinematics-Only**: This model is trained solely on the language modeling loss ($L_{text}$) to predict URDF parameters, without the [SEG] token or the associated segmentation loss.
- **Segmentation-Only**: This model is trained to generate only the link names and [SEG] tokens, optimizing exclusively for the segmentation loss ($L_{seg}$), without predicting the kinematic structure.

The results, presented in Table 5, clearly demonstrate the performance degradation when the tasks are decoupled. The **'Kinematics-Only'** model, lacking the geometric regularization provided by the segmentation task, shows a decline in parameter accuracy. This quantitatively confirms that the segmentation task acts as a crucial "booster" for learning accurate kinematic structures. More strikingly, the **'Segmentation-Only'** model also shows a drop in both mIoU and part count accuracy. This suggests that the structured reasoning required to predict the full kinematic tree forces the model to learn a more coherent and robust internal representation of the object's structure, which in turn benefits the geometric segmentation task.

These results provide strong evidence for the mutual benefits of joint prediction. Geometry provides essential regularization for predicting accurate kinematics, while the task of inferring kinematics imposes a structural prior that enhances geometric segmentation. This validates our end-to-end joint prediction paradigm as essential for achieving high-performance reconstruction.

Table 5: **Ablation on Joint Geometric and Kinematic Prediction.**

| Model Variant | Loss | Kinematic | | | Segmentation | |
|---|---|---|---|---|---|---|
| | | Type | Axis | Origin | mIoU | Count Acc |
| Kinematics-Only | $L_{text}$ | 0.009 | 0.138 | 0.175 | - | - |
| Segmentation-Only | $L_{seg}$ | - | - | - | 0.61 | 0.89 |
| **URDF-Anything (Joint)** | $L_{seg} + L_{text}$ | **0.008** | **0.132** | **0.164** | **0.63** | **0.97** |

**Summary of Design Choices.** The results from these key ablations underscore a central finding of our work: constructing a successful framework for this task is technically non-trivial. The preceding experiments demonstrate that **seemingly plausible, simpler approaches systematically fail**. For instance, relying on 2D image inputs (Table 4) leads to geometrically imprecise results, while decoupling the geometric and kinematic prediction tasks (Table 5) results in error propagation and performance degradation in both domains. Our success therefore stems not from a simple combination of modules, but from a series of deliberate, non-trivial design choices—from the foundational use of 3D point clouds to the end-to-end joint reasoning mechanism.

## 5    Conclusion

In this paper, we presented **URDF-Anything**, a novel end-to-end framework that leverages the power of 3D Multimodal Large Language Models (MLLMs) to reconstruct functional URDF digital twins of articulated objects directly from visual observations. By utilizing the MLLM's inherent capabilities and introducing a dynamic $[SEG]$ token mechanism for joint geometric segmentation and kinematic parameter prediction, our method overcomes limitations of prior decoupled or simplified approaches. Experiments demonstrate that URDF-Anything achieves state-of-the-art performance across segmentation, parameter prediction, and physical executability metrics on the PartNet-Mobility dataset, exhibiting superior generalization to complex and unseen objects. This work provides a robust and efficient solution for automated articulated object digital twin creation, significantly advancing capabilities for robotic simulation.

**Limitations.** While URDF-Anything demonstrates significant advancements, limitations include the inability to generate certain URDF properties (e.g., mass, moment of inertia), partly due to training data and base model constraints. Additionally, the pipeline is not fully end-to-end, relying on an external point-to-mesh conversion module to generate link geometry. Precision of numerical parameters is also limited by the token-based generation approach.

## Acknowledgements

This work was supported by the National Natural Science Foundation of China (62476011).

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

# A Appendix

## A.1 Dataset Preparation Details

Following prior work [9, 10], we define In-Distribution (ID) objects as those from five categories: Laptop, Box, Refrigerator, StorageFurniture, and Table. All other 41 categories in the PartNet-Mobility dataset are held out as Out-of-Distribution (OOD) to evaluate generalization to entirely unseen object classes.

The PartNet-Mobility dataset provides raw URDF files and mesh files. To prepare the data for our MLLM framework, we performed several processing steps to ensure consistency and extract necessary information.

Firstly, we performed a coordinate normalization and URDF structure regularization process. This involved establishing a consistent 'base' link as the root reference frame for each object's kinematic tree. We then programmatically calculated the transformation (position $xyz$ and orientation $rpy$) from this 'base' link to each subsequent link's frame. The URDF structure was then modified: joints not already having 'base' as their parent were re-configured so that their child link connects directly to the 'base' link. The '<origin>' tag of these re-parented joints was updated to the previously calculated transformation from 'base' to the child link's frame, effectively defining the child link's pose relative to the 'base'. Additionally, for links that originally referenced multiple mesh files within their '<visual>' and/or '<collision>' tags (e.g., multiple '<visual>' elements each with a mesh), these URDF entries were consolidated. Each link was modified to reference a single, representative mesh file for its visual geometry and similarly for its collision geometry. The original local transform (origin $xyz$ and $rpy$) specified in the first-found visual or collision mesh entry for that link was preserved and applied to these new consolidated entries. This overall simplification aimed to streamline subsequent geometric processing. An example of an original URDF file, which serves as input to our processing, is shown in Figure 8. The result of this URDF processing, particularly the consolidated representation for links, is illustrated in Figure 8.

Secondly, for the purpose of dataset splitting and ensuring manageable complexity during training, we filtered the dataset based on the number of parts. Specifically, objects with fewer than 8 articulated parts (links involved in joints) were selected to form the primary dataset used for training and evaluation splits.

Finally, since our framework requires visual input, we generated synthetic multi-view and multi-state RGB images from the processed 3D models using the SAPIENS simulator. For multi-view rendering, we employed two strategies: capturing viewpoints from an equator plane and capturing viewpoints distributed spherically (Figure 4). The spherical viewpoint distribution utilized a minimum potential energy method to ensure relatively uniform coverage around the object. These rendered images, along with the corresponding point clouds derived from the 3D models, served as the visual input for our model.

## A.2 Point Cloud Generation Details

As detailed in Section3.2 the initial step of our pipeline is the generation of a dense 3D point cloud $P_{obj}$ from the input visual observations. This process adapts based on whether multi-view or single-view images are available. The generated point cloud serves as the primary geometric input for the subsequent multimodal processing stages.

### A.2.1 Multi-view Point Cloud Generation

When multiple RGB images of the articulated object from different viewpoints are available, we leverage the capabilities of the pre-trained DUSt3R model [37]. DUSt3R is specifically designed for establishing dense 2D-to-3D correspondences across multiple images and reconstructing accurate 3D geometry. We feed the set of multi-view images into DUSt3R, which outputs a dense point cloud $P_{obj}$ representing the reconstructed 3D structure of the object. Example results are showed in Figure 6.

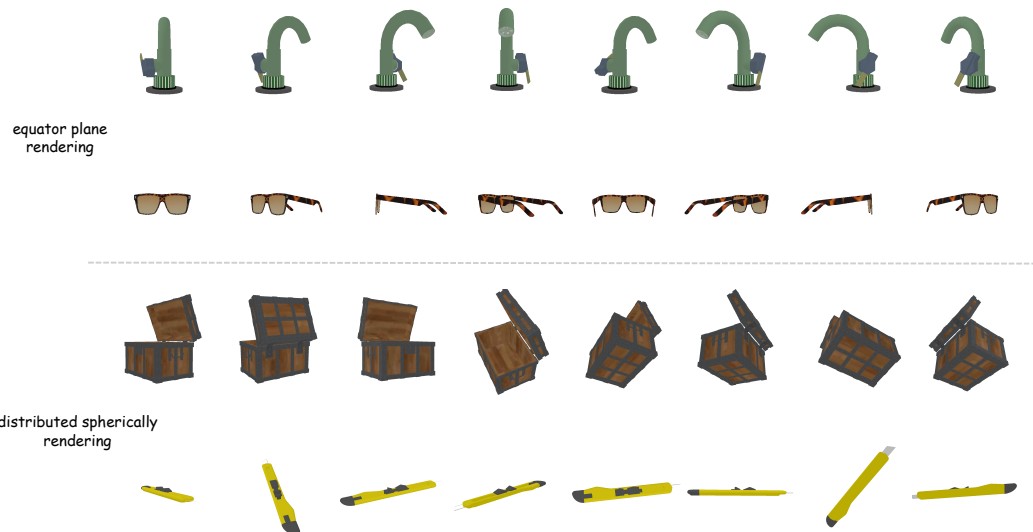

equator plane
rendering

distributed spherically
rendering

Figure 4: SAPIENS Simulator Rendering Strategies

### A.2.2 Single-view Point Cloud Generation

For the more challenging scenario where only a single RGB image of the object is provided, we employ a generative approach utilizing LGM (Large Generative Model) [38]. This process involves two main steps. First, we use a pre-trained diffusion model to synthesize a consistent set of multi-view images that are plausible renderings of the object from viewpoints around the original single image. Second, we reconstruct the 3D geometry from these synthesized multi-view images. This reconstruction can be initially represented as a 3D Gaussian Splatting model, which is then converted into the final dense point cloud representation $P_{obj}$. Example results are shown in Figure 5.

Regardless of whether the point cloud is generated from multi-view images via DUSt3R or from a single view via LGM and diffusion-based synthesis, the output of this stage is a dense point cloud $P_{obj} \in \mathbb{R}^{N \times 6}$, where each point is represented by its 3D spatial coordinates and RGB color value (XYZRGB). It is important to reiterate that this generated point cloud $P_{obj}$ provides a holistic, monolithic representation of the entire object geometry; it does not intrinsically contain part-level segmentation information.

### A.3 3D MLLM's input and output Design Details

**Input Prompt Design**

The input to our 3D MLLM consists of processed 3D point cloud features (passed to the model via a designated token, e.g., `<point>`) and a text instruction. To enhance the robustness and generalization capability of the model, we utilize several different text instruction templates during training. These templates vary in the level of detail provided about the object and its parts (e.g., including object category, number of parts, detailed description of objects) but consistently specify the desired task: predict joint parameters and segmentation masks in a structured format. A portion of these templates include the "number of parts," primarily serving as a pedagogical tool or a strong supervisory signal to ease the fitting pressure and help the model learn the correlation between an object's visual form and its internal articulated structure more effectively.

A representative example of a general input instruction template structure is:

```
The articulated object [Object Category] consists of [Number of Parts]
parts. [Optional descriptive phrases about links or overall object,
e.g., This object is a kitchen faucet. There are two distinct handles
or witches positioned on either side of the base.] Predict all joint
parameters in JSON format, including type, origin,axis, parent, and
```

```
child. Segment each link in JSON format.
```

An instance of a concrete input prompt generated from a template for a specific object (a Faucet in this case) used in our experiments is shown below. Note that the full input to the MLLM would include the point cloud features encoded prior to this text sequence. During Inference, providing the number of parts is entirely optional. For all quantitative evaluations in the paper and for fair comparison with other models, we uniformly used a generic prompt that did not contain any information about the number of parts or joints (e.g., "Please segment the object and predict its joint parameters").

```
This articulated object Faucet consists of 4 parts. Predict all joint
parameters in JSON format, including type, origin, axis, parent, and
child. Segment each link in JSON format.
```

**Output Format Design**

The MLLM is trained to autoregressively generate its output as structured text in JSON format. This output design is critical as it simultaneously specifies the predicted kinematic structure and provides explicit signals for geometric segmentation. The output JSON contains two primary keys:

- `"joints"`: This key maps to a list, where each element represents a predicted joint. Each joint object includes the essential URDF parameters: `"id"`, `"type"` (e.g., "revolute", "prismatic", "fixed"), `"parent"` link name, `"child"` link name, `"origin"` (containing `"xyz"` position and `"rpy"` orientation), and `"axis"` vector.

- `"links"`: This key maps to an object that provides information about the predicted links. Specifically, it maps each predicted link name (e.g., "link_0", "link_1") to the predicted semantic category of the part followed immediately by the special [SEG] token (e.g., `"switch[SEG]"`). The presence and position of the [SEG] token in the output sequence is the explicit signal generated by the MLLM that triggers and guides the geometric segmentation process for the corresponding link in the input point cloud (as detailed in Section 3.4 of the main paper).

A complete example of the MLLM's generated output in JSON format for the Faucet instance, corresponding to the input prompt shown above, is provided below:

```
{
    "joints": [
        {
            "id": "joint_0",
            "type": "revolute",
            "parent": "base",
            "child": "link_0",
            "origin": {
                "xyz": [-0.079, -0.48747, -0.0],
                "rpy": [1.5708, -0.0, 1.5708]
            },
            "axis": [0.0, 1.0, 0.0],
            "limit": {"lower": 0, "upper": 1.57}
        },
        {
            "id": "joint_1",
            "type": "revolute",
            "parent": "base",
            "child": "link_1",
            "origin": {
                "xyz": [-0.079, 0.49568, -0.0],
                "rpy": [1.5708, -0.0, 1.5708]
            },
            "axis": [0.0, -1.0, 0.0],
            "limit": {"lower": 0, "upper": 1.57}
        },
```

```
        {
            "id": "joint_2",
            "type": "continuous",
            "parent": "base",
            "child": "link_2",
            "origin": {
                "xyz": [-0.079, 0.00411, -0.0],
                "rpy": [1.5708, -0.0, 1.5708]
            },
            "axis": [0.0, 1.0, 0.0]
        },
        {
            "id": "joint_3",
            "type": "fixed",
            "parent": "base",
            "child": "link_3",
            "origin": {
                "xyz": [0.0, 0.0, 0.0],
                "rpy": [1.5708, 0.0, 1.5708]
            },
            "axis": [1.0, 0.0, 0.0]
        }
    ],
    "links": {
        "link_0": "switch[SEG]",
        "link_1": "switch[SEG]",
        "link_2": "spout[SEG]",
        "link_3": "faucet_base[SEG]"
    }
}
```

## A.4  Shape Reconstruction Quality

To quantitatively evaluate the geometric fidelity of the final mesh outputs—including the point-to-mesh conversion stage—we compute the Chamfer Distance (CD) between our generated meshes and the ground-truth meshes from PartNet-Mobility. We compare against two strong baselines that also produce explicit mesh outputs. The results, shown in Table 6, clearly demonstrate the superiority of our method in shape reconstruction quality.

Table 6: **Comparison of Shape Reconstruction Quality (Chamfer Distance)**.

| Method | ALL | ID | OOD |
|---|---|---|---|
| CARTO | 1.24 | 0.88 | 1.27 |
| PARIS | 3.06 | 2.17 | 3.13 |
| **URDF-Anything (Ours)** | **1.39** | **0.40** | **1.51** |

## A.5  Failure Case Analysis

We analyze failure cases from the physical executability evaluation (22% failure rate overall). As shown in Table 7, the vast majority of failures (21%) stem from incorrect kinematic parameters—such as misaligned joint origins or axes—leading to non-physical motion (e.g., parts colliding or detaching). Only 1% of failures are due to invalid JSON format, confirming our MLLM reliably generates valid syntax. To provide a more detailed error analysis for physical executability, we have conducted a deeper failure analysis, shown in Table 8.

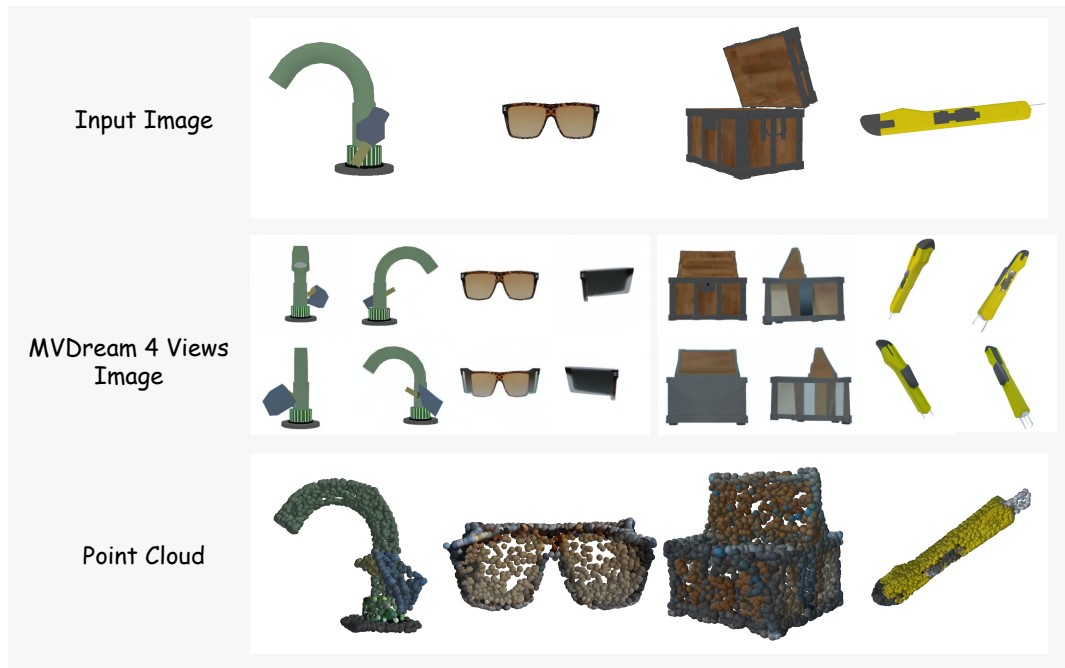

Figure 5: LGM: Point Cloud Generation via Multi-view Synthesis

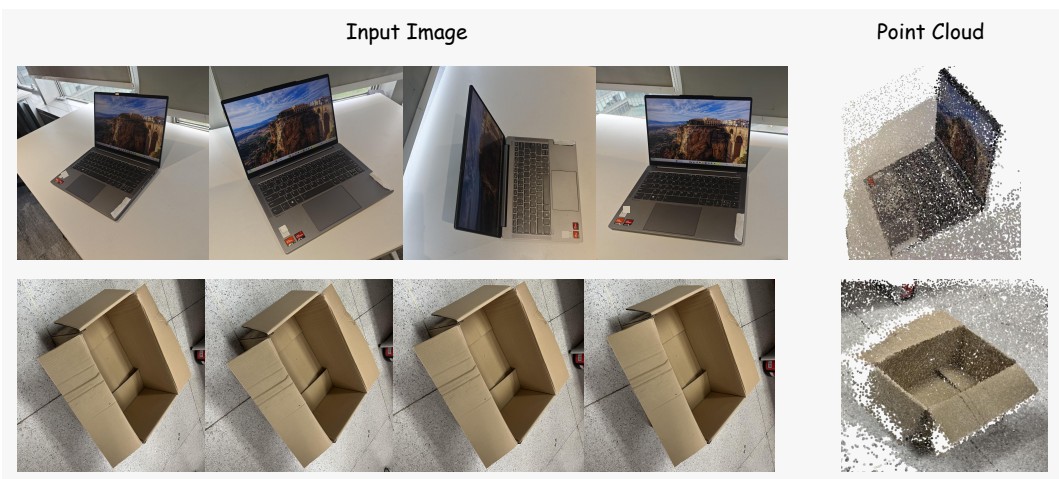

Figure 6: Multi-view Point Cloud Generation using DUSt3R.

## A.6 Comparison with Alternative Methodologies

For completeness, this appendix provides a targeted comparison against alternative methods like CARTO [30] and PARIS [32], which address more constrained problems under different assumptions.

- **PARIS** [32] is an optimization-based method requiring images of start/end states. Its per-instance optimization is slow at inference (>3min), making it impractical for many applications.
- **CARTO** [30] is a fast feed-forward model, but it is highly specialized for single-joint objects and cannot perform part segmentation or generate a complete, executable URDF.

Tables 9 and 10 quantify the trade-offs in speed, accuracy, and capability. While CARTO is faster, its simplicity results in high prediction errors and extremely. PARIS's optimization-based approach also struggles to find physically plausible solutions, leading to poor accuracy and executability (25%).

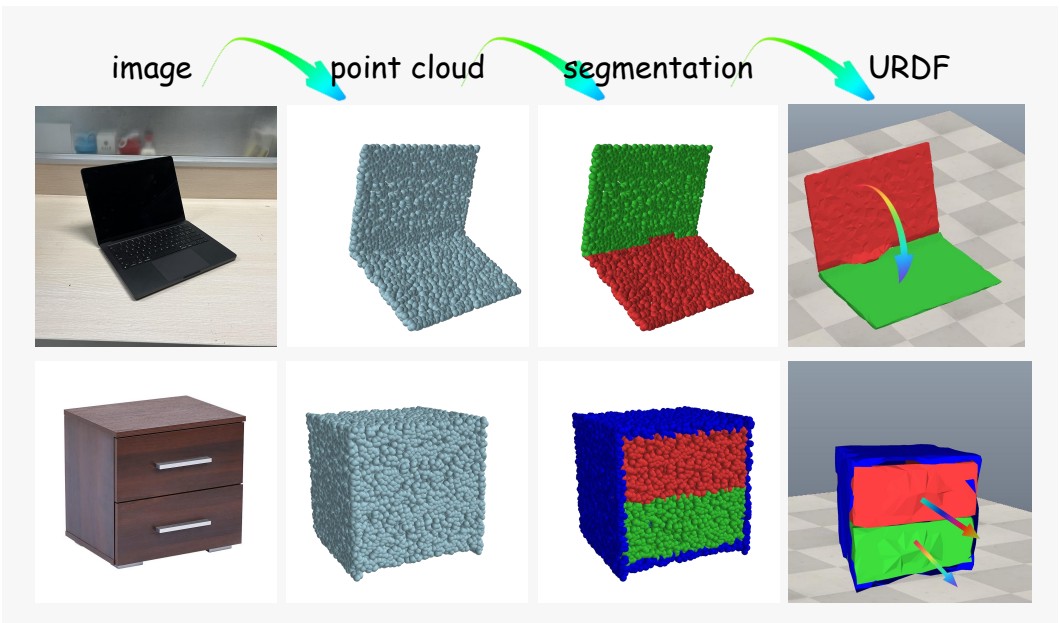

Figure 7: Step-by-Step Reconstruction from Real-World Image to Functional URDF.

Table 7: **Breakdown of Physical Executability Failures.**

| Failure Type | Percentage |
|---|---|
| Incorrect joint parameters | 21% |
| JSON Format Error | 1% |
| **Overall Failure** | 22% |

In contrast, URDF-Anything's MLLM-based design provides a robust balance, achieving vastly superior accuracy and a high executability rate (90%) with a practical feed-forward inference time ( 13s). This result validates our design choice for tackling complex, general-purpose reconstruction.

### A.7 Zero-Shot Sim-to-Real Generalization Evaluation

To assess the real-world applicability of our framework, we conducted a zero-shot sim-to-real evaluation. Our model, trained exclusively on the simulated PartNet-Mobility dataset, was tested directly on the real-world portion of the PARIS dataset [32]. This dataset provides real-world captures and annotations for two main categories: Fridge and Storage.

The quantitative results of this evaluation are presented in Table 11.

These results provide several key insights into the model's sim-to-real transfer capabilities. First, our method achieves reasonable performance on geometric tasks (segmentation mIoU and mesh CD) and continuous parameter prediction (axis/origin), despite the significant domain gap between synthetic training data and real-world images. For the Fridge and Storage categories present in the dataset, the model correctly identified the joint type in all cases.

### A.8 More Ablation

**Impact of Context Fusion in Segmentation.** We also analyzed the impact of our proposed context fusion mechanism for the [SEG] token. We compared our approach, which fuses the hidden states of the [SEG] token and its preceding category token, against a baseline that uses only the generic hidden state of the [SEG] token ($h_{seg}$). Table 12 shows that our context fusion design achieves a clear improvement in segmentation mIoU, confirming its effectiveness in providing part-aware context for fine-grained segmentation.

```xml
<?xml version="1.0" ?>
<robot name="UNNAMED_34">
 <!-- base link --!>
 <link name="base"/>
 <link name="link_0">
  <visual>
   <origin rpy="0.0 0.0 0.0" xyz="0.0 0.0 0.0"/>
   <geometry>
    <mesh filename="link_0_combined_mesh.obj"/>
   </geometry>
  </visual>
  <collision>
   <origin rpy="0.0 0.0 0.0" xyz="0.0 0.0 0.0"/>
   <geometry>
    <mesh filename="link_0_combined_mesh.obj"/>
   </geometry>
  </collision>
 </link>
 <!-- link 1 --!>
 <link name="link_1">
  <visual>
   <origin rpy="0.0 0.0 0.0" xyz="-0.67 0.22 0.16"/>
   <geometry>
    <mesh filename="link_1_combined_mesh.obj"/>
   </geometry>
  </visual>
  <collision>
   <origin rpy="0.0 0.0 0.0" xyz="-0.67 0.220.16"/>
   <geometry>
    <mesh filename="link_1_combined_mesh.obj"/>
   </geometry>
  </collision>
 </link>
 <!-- joint 0 --!>
 <joint type="fixed" name="joint_0">
  <parent link="base"/>
  <child link="link_0"/>
  <origin rpy="1.57 0.0 1.57" xyz="0.0 0.0 0.0"/>
  <axis xyz="1 0 0"/>
 </joint>
 <!-- joint 1 --!>
 <joint type="revolute" name="joint_1">
  <parent link="base"/>
  <child link="link_1"/>
  <origin rpy="1.57 -0.0 1.57" xyz="-0.16 0.67 -0.22"/>
  <axis xyz="-1.0 0.0 0.0"/>
  <limit lower="-1.83" upper="-0.0" effort="0.0" velocity="0.0"/>
 </joint>
</robot>
```

Figure 8: Example Generated URDF File

Table 8: **Breakdown of Physical Executability Failures by Category.**

| Category | Total (%) | JSON Format Error (%) | Joint Error (%) | | |
| --- | --- | --- | --- | --- | --- |
| | | | Type | Axis | Origin |
| Window | 15.5 | 0.00 | 0.03 | 0.05 | 0.07 |
| Chair | 22.2 | 0.01 | 0.06 | 0.06 | 0.09 |
| Globe | 14.8 | 0.00 | 0.03 | 0.05 | 0.07 |

Table 9: **Comparison of Inference Speed and Methodology.**

| Method | Core Methodology | Avg. Inference Time |
| --- | --- | --- |
| CARTO [30] | Feed-forward Encoder-Decoder | 1s |
| PARIS [32] | Per-instance Optimization | >3min |
| **URDF-Anything (Ours)** | Feed-forward MLLM Inference | 13s |

Table 10: **Comparison of Reconstruction Accuracy and Physical Executability.** Our method delivers substantially lower prediction errors and a much higher rate of generating functional URDFs.

| Method | mIoU | CD | Type Error | Axis Error | Origin Error | Executability |
| --- | --- | --- | --- | --- | --- | --- |
| CARTO [30] | - | **1.24** | 0.12 | - | - | - |
| PARIS [32] | 0.44 | 3.06 | 0.25 | 0.84 | 0.30 | 25% |
| **URDF-Anything(Ours)** | **0.69** | 1.39 | **0.007** | **0.121** | **0.13** | **90%** |

Table 11: **Zero-Shot Sim-to-Real Performance on the PARIS Real-World Dataset.** The model was trained only on simulation data and tested on real-world images.

| Category | mIoU | CD | Type Error | Axis Error | Origin Error |
| --- | --- | --- | --- | --- | --- |
| Fridge | 0.57 | 1.03 | 0.0 | 0.335 | 0.256 |
| Storage | 0.56 | 0.99 | 0.0 | 0.362 | 0.349 |

Table 12: **Ablation: Context Fusion Mechanism for Segmentation.**

| Segmentation Mechanism | Query Feature | mIoU |
| --- | --- | --- |
| Generic only | $h_{seg}$ | 0.58 |
| **Context Fusion (Ours)** | $h_{combined} = [h_{category}; h_{seg}]$ | **0.63** |

## A.9 Analysis of the Joint Segmentation and Kinematics Learning Mechanism

A critical aspect of our framework is the tight coupling between geometric segmentation and kinematic parameter prediction, facilitated by the '[SEG]' token and end-to-end joint optimization. This section aims to demystify this process, explaining it as a form of "Geometric Regularization" and providing empirical validation.

**Theoretical Explanation: Geometric Regularization via Joint Optimization**   The core principle behind this synergy is that the segmentation task ($L_{seg}$) provides a powerful physical anchor for the kinematic prediction task ($L_{text}$). A model predicting only kinematics might hallucinate a joint that is textually plausible but physically baseless. Our joint optimization forces the model to ground its abstract kinematic predictions in the concrete geometric entities of the point cloud. This geometric grounding effectively constrains the parameter search space, guiding the model toward more physically consistent articulation structures. This is enabled by the shared representation in our end-to-end model, which forces the hidden states used for both tasks to be mutually informative.

**Experimental Validation**    To empirically validate this theory, we provide both quantitative and qualitative evidence.

**Quantitative Ablation.**  We trained a variant of our model without the '[SEG]' token and its associated segmentation loss ($L_{seg}$). As shown in Table 5, 'Kinematics-Only' removes the geometric regularization from the segmentation task, leading to a consistent increase in prediction error across all kinematic parameters. This quantitatively demonstrates that joint segmentation is crucial for achieving high-fidelity kinematic inference.

**Qualitative Visualization.** To provide intuitive insight into the "black-box" mechanism, we visualized the self-attention maps from a key transformer layer while the model generated a kinematic parameter token (e.g., 'axis'). As illustrated in Figure 9, the full model's attention is sharply focused on the physically relevant joint region (the hinge). In contrast, the model trained without segmentation exhibits diffuse, unfocused attention. This visualization provides direct evidence that our joint optimization strategy forces the model to ground abstract kinematic concepts in concrete geometric features, demystifying the learning process.

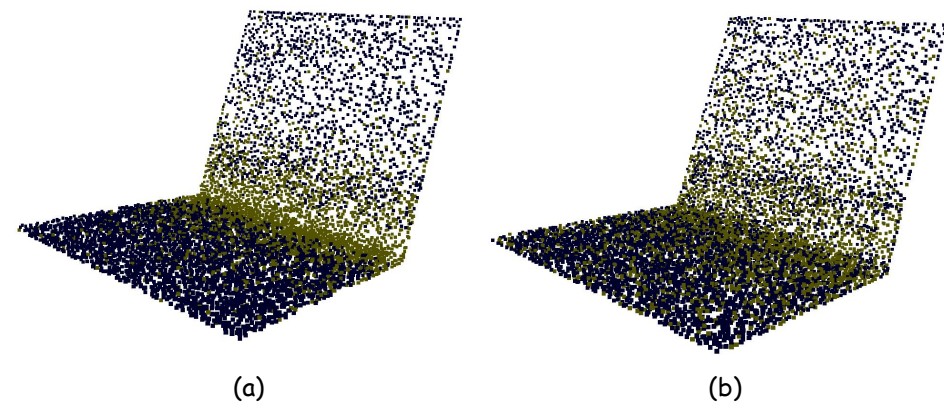

(a)               (b)

Figure 9: **Attention Visualization for Kinematic Token Generation.** (a) Our full model, trained with joint segmentation, concentrates its attention (yellow areas) on the relevant joint region when predicting the joint 'axis'. (b) The model trained without the segmentation loss shows diffuse attention, lacking precise geometric grounding.

