# OpenReview forum: "URDF-Anything: Constructing Articulated Objects with 3D Multimodal Language Model"
_NeurIPS.cc/2025/Conference — NeurIPS 2025 spotlight_

### Official Review · Reviewer_7j6V · 2025-06-02

**Clarity:** 3
**Significance:** 3
**Originality:** 2
**Rating:** 3
**Confidence:** 4

**Summary:**

The paper presents URDF-Anything, an end-to-end pipeline that turns RGB observations (single- or multi-view) into functional URDF parameters of articulated objects. A 3D multimodal LLM (ShapeLLM-7B + LoRA fine-tuning) jointly (i) segments a dense point-cloud into links and (ii) predicts joint parameters. The key technical novelty is a dynamic [SEG] token that the model inserts while autoregressively generating the JSON-style URDF: each [SEG] token’s hidden state is used, via cross-attention, as a query to produce a per-point mask for the corresponding link.

**Questions:**

See above

**Ethical Concerns:**

["NO or VERY MINOR ethics concerns only"]

**Final Justification:**

revised score

**Limitations:**

It would be better to show some failure cases.

**Quality:**

3

**Strengths And Weaknesses:**

**Strengths**
- The paper is well-written.
- To my knowledge, the first work to leverage a 3D MLLM with explicit token-based cross-modal segmentation for simultaneous geometry + kinematics prediction; elegant integration of language generation and point-level supervision.
- Comprehensive pipeline—from point-cloud creation (DUSt3R / LGM) through [SEG] cross-attention to mesh fusion—explained clearly.
- Strong quantitative gains on both ID and OOD splits; executability metric moves beyond static accuracy; ablations on modality and [SEG] tokens are informative.
- Produces ready-to-simulate URDFs, a strong link to robotics/embodied AI workflows.

**Weaknesses**
- Results lack error bars or significance tests; variance is only given for one table.
- The pipeline still relies on an external point-to-mesh post-process; errors there are not evaluated.
- Training on ShapeLLM-7B with LoRA is resource-heavy; training time is not reported.
- Mass, inertia, friction and joint damping are omitted; executability counts loading & actuation but not quantitative dynamics accuracy.

---

> ### Author Rebuttal · Authors · 2025-07-31
>
> We sincerely thank the reviewer for their encouraging and positive assessment. We are especially grateful for your acknowledgment of the paper's clarity, the novelty of our technical approach in using a 3D MLLM for joint prediction, and the elegance of the integration. Furthermore, we appreciate your recognition of our strong quantitative results, particularly on OOD data, and the practical value of our ready-to-simulate URDF outputs for the robotics community.
>
> ## W1: Lack of Error Bars
>
> We sincerely thank the reviewer for this constructive feedback. We agree that providing a thorough analysis of result variance is crucial for demonstrating the robustness of our method.
>
> We have now calculated the standard deviations for our key metrics, which we are happy to provide for clarification.
>
> Table 1: Quantitative Results for Part-Level Link Segmentation.
>
> | Models | mIoU ALL ↑ | mIoU ID ↑ | mIoU OOD ↑ | Count Acc ALL ↑ | Count Acc ID ↑ | Count Acc OOD ↑ |
> | :--- | :---: | :---: | :---: | :---: | :---: | :---: |
> | Uni3D w/o text | 0.36±0.01 | 0.50±0.02 | 0.33±0.01 | 0.73±0.02 | 0.83±0.03 | 0.70±0.01 |
> | Uni3D w/ text | 0.54±0.02 | 0.64±0.01 | 0.51±0.02 | 0.84±0.02 | 0.91±0.02 | 0.82±0.02 |
> | **URDF-Anything ([SEG])** | **0.63**(16.7%↑)±0.01 | **0.69**±0.01 | **0.62**±0.01 | **0.97**(15.4%↑)±0.02 | **0.99**±0.02 | **0.96**±0.02 |
>
> Table 3: Physical Executability Rate (%↑).
>
> | Method | ALL | ID Classes (Simple) | ID Classes (Complex) | OOD Classes (Simple) | OOD Classes (Complex) |
> | :--- | :---: | :---: | :---: | :---: | :---: |
> | URDFormer Oracle | 24%±8.24% | 40%±4.53% | 23%±5.30% | 27%±5.67% | 11%±11.62% |
> | Articulate-Anything | 52%±12.63% | 71%±6.24% | 42%±9.49% | 69%±11.70% | 35%±14.72% |
> | **URDF-Anything(Ours)** | **78%**(50.0%↑)±1.51%  | **90%** ±1.20% | **78%** ±0.87% | **83%** ±1.56% | **67%** ±1.58% |
>
> Table 4: Ablation Study on Input Modality for Joint Parameter Prediction.
>
> | Method Variant | Input Modality | Type ↓ | Axis ↓ | Origin ↓ |
> | :--- | :--- | :---: | :---: | :---: |
> | OBB | Text | 0.42±0.18 | 0.70±0.26 | 0.47±0.22 |
> | Point Cloud only | Point Cloud | 0.34±0.11 | 0.29±0.18 | 0.26±0.15 |
> | Qwen2.5-VL-7B | Image + Text | 0.57±0.14 | 0.85±0.31 | 0.23±0.16 |
> | Qwen2.5-VL-7B + ft | Image + Text | 0.38±0.09 | 0.81±0.24 | 0.18±0.10 |
> | **Point Cloud + Text (Ours)** | Point Cloud + Text | **0.008**±0.001 | **0.132**±0.486 | **0.164**±0.260 |
>
> We will incorporate these updated tables with full variance reporting into the revision of our paper. Thank you again for your valuable suggestion, which has helped improve the rigor of our work.
>
> ## W2: Post-processing Step
>
> Thank you for highlighting this important evaluation aspect. To address your point, we have conducted a new experiment to quantitatively evaluate the final mesh output from our pipeline, including the point-to-mesh post-processing stage.
> We measured the geometric fidelity using the standard **Chamfer Distance (CD) metric**. Results are as follows. We will add this evaluation to the revision.
>
> | Method | ALL | ID Classes | OOD Classes |
> | :- | :-: | :-: | :-: |
> | Ours | 1.39 | 0.40 | 1.51 |
>
>
>
> ## W3: Unreported Training Resources
>
> Thank you for pointing out this omission. Our model was fine-tuned in
> **2.5 hours on a single NVIDIA A800 (80GB) GPU**. We will add these details to the revision of the paper and make the code open source.
>
>
> ## W4: Lack of Dynamic Parameters
>
> We thank the reviewer for this insightful question, which allows us to clarify the well-defined scope of our work. Our primary focus is on a central challenge in robotic manipulation: reconstructing an object's **geometry and kinematic structure** from visual input.
>
> This focus **is consistent with the established scope** of other state-of-the-art methods in this domain. Prominent works, including Articulate-Anything, Real2Code, and URDFormer, also concentrate exclusively on geometric and kinematic parameters. The estimation of dynamic properties (mass, inertia, friction, etc.) from vision alone is a distinct and highly challenging research problem, which the community typically treats separately.
>
> We argue that providing a kinematically accurate model is, in itself, a highly valuable contribution. An accurate kinematic model alone is sufficient to **meet the requirements of a vast range of robotic applications**, including manipulation planning, grasping, and task simulation. In many real-world scenarios, a robot's own controllers (e.g., through impedance or admittance control) **can adapt to an object's unknown dynamics** during interaction, making the kinematic model the most critical component for planning and execution.
>
> Therefore, our contribution is a robust and complete solution to the core kinematic reconstruction problem. While incorporating dynamics is an exciting future direction, our work establishes the essential structural foundation upon which such dynamic models could eventually be built.
>
> ## Limitations: Failure Case Analysis
>
> Thank you for this valuable suggestion. We will add a dedicated section to the appendix in our revision to include and discuss visualizations of typical failure cases.

---

> > ### Comment · Reviewer_7j6V · 2025-08-08
> >
> > Thanks for the rebuttal. I will keep my score.

---

### Official Review · Reviewer_qimd · 2025-06-11

**Clarity:** 2
**Significance:** 3
**Originality:** 3
**Rating:** 5
**Confidence:** 5

**Summary:**

This paper introduces URDF-Anything, a framework for reconstructing articulated 3D objects as URDF models directly from visual input, using a 3D Multimodal Large Language Model (MLLM). The authors first convert single-view or multi-view RGBs to point-cloud via off-the-shelf DUStr and LGM. Then, they train a 3D multimodal Large Language Model (3D MLLM) to take the point-cloud and some text prompt to perform part segmentation, link hierarchy and joint prediction.

**Questions:**

- How is object category (e.g., "faucet") and number of parts determined at inference time? Can your model infer this from the point cloud, or does it rely on annotations (as in PartNet-Mobility)?
- Please provide a breakdown of performance by object category (e.g., cabinet, laptop, faucet) and a failure case analysis. This would help assess robustness and generalization.

**Ethical Concerns:**

["NO or VERY MINOR ethics concerns only"]

**Final Justification:**

The authors have addressed all my concerns during the rebuttal

**Limitations:**

yes

**Quality:**

2

**Strengths And Weaknesses:**

**Strengths**

- The idea of using a 3D MLLM for articulation is novel and interesting.
- Ablation study 4.4 provides valuable insight, highlighting the benefits of explicit 3D input over 2D image-based alternatives.
- The experimental results show gains in segmentation, joint prediction accuracy, and simulation executability.

**Weaknesses**
- The related work section is imprecise and at times inaccurate.
	- "Research on [...] is a related but distinct field." is not accurate and confusing. Many works under this section are directly relevant to this work. In fact, the authors themselves compare against these methods as baselines.
	- "[29] focuses on generating articulated simulation scenes from images, which is related but distinct from accurate individual object" is not accurate. URDFormer can also generate individual objects. In fact, the authors did compare against this baseline.
- The point cloud generation pipeline is unnecessarily complicated. The authors use DUSt3R for multi-view input and LGM for single-view, but don’t explain why a unified approach (e.g., always using LGM or DUSt3R after synthesized views) isn’t employed. This suggests convenience-driven decisions rather than principled design.
	- For example, they could hallucinate single-view to multi-view via off-the-shelf model (ImageDream) then use DUSt3R. Alternatively, they can also scrape off DUSt3R and just use multi-view posed RGBs directly with LGM.
- Some important aspects of the system’s training and inference remain ambiguous:
	- The "Uni3D w/ text" baseline is underspecified. What exact text input is used? The generic prompt "Segment the input point cloud into parts and predict articulation parameters." shown in Fig. 2 seems insufficient for meaningful conditioning.
	- It is unclear how "Uni3D w/o text" and "Uni3D w/ text" perform segmentation. Do they involve a classifier head trained on predefined categories? If I understand correctly, in the Uni3D paper, they use contrastive-style loss to do segmentation. So during inference, they likely provided a list of semantic part names that need to be segmented.
	- A more intuitive and technical explanation is needed to justify why the [SEG] mechanism outperforms "Uni3D w/ text", especially since both involve feature-text fusion at different levels.
Evaluation choices raise some concerns:
	- Line 208 defines a custom “joint success” criterion but omits reporting success rate as defined in prior work [1]. Why?
	- Table 3 (physical executability) omits Real2Code, even though it's included in other comparisons.

---

> ### Author Rebuttal · Authors · 2025-07-31
>
> We sincerely thank the reviewer for their positive feedback. We are particularly encouraged by your recognition of the novelty of our core idea, the valuable insights from our ablation study, and the strong experimental gains across segmentation, joint prediction, and simulation executability.
>
> ## W1: Related Work Section
>
> Thank for pointing out the imprecisions in our Related Work section. We will make the following corrections:
>
> Rewrite "Research on [...] is a related but distinct field." to "The automated modeling of articulated objects for robotic manipulation is an active field of research, with several distinct methodological approaches."
>
> Rewrite "[29] focuses on generating articulated simulation scenes from  images, which is related but distinct from accurate individual object" to "[29] is fundamentally limited by its reliance on a hard-coded system to assign kinematic parameters and retrieve meshes based on the network's discrete part classification output, a method which compromises final geometric and kinematic fidelity."
>
> ## W2: Input Pipeline
>
> Thanks for in-depth thinking on our input pipeline design. We fully agree that the design should be principled rather than based on convenience.
>
> Our core principle is to use the most suitable, highest-fidelity method available for each type and quality of input source. This ensures that the geometric information received by our core model is of the highest possible quality. The two input scenarios (real multi-view vs. single-view) are fundamentally different problems in nature and difficulty:
>
> **For multi-view input (using DUSt3R)**: When real, high-quality multi-view images are available, they contain the most reliable geometric information. In this case, the most direct and high-fidelity approach is to use a specialized multi-view 3D reconstruction algorithm. We chose DUSt3R because it is a classical method in this specific domain (reconstructing geometry from real multi-view images). Forcing the use of a generative method like LGM in this scenario would actually degrade the quality of the point cloud fed to our core model by introducing artifacts and uncertainties. It is unnecessary to "downgrade" real data to generated data before processing.
>
> **For single-view input (using LGM)**: When only a single-view image is available, the problem fundamentally changes from 'reconstruction' to 'conditional generation' or 'hallucination'. We must leverage a generative model to complete the missing 3D information. We chose LGM because it is a classical solution for this specific problem (generating 3D from a single view).
>
> Regarding your suggestions for a unified pipeline:
>
> **On using "ImageDream + DUSt3R"**: This insight actually points to the internal workings of models like LGM. Advanced single-view generative models such as LGM already integrate modules similar to MVDream/ImageDream for synthesizing multiple views from a single one. Therefore, using the highly integrated and optimized LGM is a more rational and efficient choice than manually assembling a similar pipeline.
> **On using "LGM in the multi-view case as well"**: As mentioned before, this would mean applying a tool designed for "generation" to a "reconstruction" problem, which is not the optimal choice.
>
>
> ## W3
>
> ### W3.1 & W3.2: Uni3D Baselines / Uni3D
>
> Here we clarify the implementation and purpose of our two key baselines:
> **‘Uni3D w/ text’** simulates text-guided segmentation. Following the Uni3D paper, we freeze the backbone and train a feature propagation layer to align local point features with text features from a provided list of part names (e.g., ['furniture body', 'laptop base']). At inference, each point is assigned the label of the text prompt with the highest feature similarity.
>
> **‘Uni3D w/o text’** is a supervised closed-set method designed to test the raw quality of the pre-trained geometric features. Inspired by classic methods like PointNet, we add an MLP head to the frozen Uni3D backbone and train only this head to classify points into a predefined set of all part categories from complete dataset(including both ID and OOD objects). This process requires no text input.
>
> The purpose of these baselines is twofold:
> The poor performance of ‘Uni3D w/o text’ demonstrates that relying on purely geometric features is insufficient and proves the necessity of semantic guidance for this task. Its weakness stems from a paradigm mismatch, as it discards the rich, cross-modal semantic alignment capability Uni3D was pre-trained for.
>
> By outperforming ‘Uni3D w/ text’, we show that our [SEG] token mechanism is a superior guidance strategy. Unlike static text prompting, our method dynamically and end-to-end couples the segmentation signal with the autoregressive generation of the kinematic structure, leading to more accurate and context-aware results.
>
> ### W3.3: Justification for [SEG]'s Superiority
>
> The superiority of our [SEG] token mechanism over the baseline stems from two fundamental differences:
> First, our [SEG] token is a **dynamic and context-rich signal**, unlike a static word used in the ‘Uni3D w/ text’ baseline. It is generated by the LLM after the model has already processed both the **complete point cloud geometry** and the **preceding kinematic parameters**(e.g., parent-child links, joint types) within the autoregressive sequence. Therefore, the token’s embedding is inherently informed by both geometric and structural context, making its guidance for segmentation far more precise than a simple class label.
> Second, our training paradigm **learns a joint probability distribution over both kinematic structure and geometry**. Through end-to-end optimization, the generation of a plausible structure and the successful segmentation of its corresponding geometry are tightly coupled. This is fundamentally different from the baseline's decoupled approach, where segmentation is merely a post-hoc classification task. In essence, our method transforms segmentation from simple matching into a context-aware grounding problem, leading to superior accuracy.
> To prove the crucial role of our [SEG] token mechanism, we have conducted new ablation studies. We summarize the key experiments and findings below:
> 1. **Segmentation-Only**: This model only outputs [SEG] tokens, trained solely for part segmentation and does not generate any kinematic structure. This experiment validates the necessity of "context" and "joint training".
> 2. **Geometry-Blind**: In this model, the LLM is given ground-truth kinematic structure but has no access to the input point cloud when attempting to output segmentation commands. This experiment validates the necessity of "geometric features" for the [SEG] token.
>
> Results are as follows:
>
> | Model | Segmentation Accuracy (mIoU ↑) | Count Accuracy (Count Acc ↑) |
> | :-: | :-: | :-: |
> | Uni3D w/ text | 0.54 | 0.84 |
> | Our Full Model | 0.63 | 0.97 |
> | | | |
> | Ablation 1: Segmentation-Only | 0.61 | 0.96 |
> | Ablation 2: Geometry-Blind | ~0.1 (Near-random) | - |
>
> The performance of the "Segmentation-Only" model is slightly lower than that of the full model, indicating that kinematic structural context and joint structure-geometry training are beneficial for achieving high-precision segmentation.
> The "Geometry-Blind" model completely fails the segmentation task, decisively proving that the **LLM's direct perception of geometric features** is the cornerstone of the [SEG] mechanism's functionality.
>
> ### W3.4: Omission of Standard "Joint Success Rate" Metric
>
> Thanks for pointing this out. Due to the length constraints of rebuttal, please refer to Table 1 in our response to reviewer KtPh, for a complete breakdown of the joint success rate.
>
> ### W3.5: Omission of Real2Code in Table 3 / Table
>
> The reason for its initial absence is that the official LLM model checkpoint and training code for Real2Code have not been released. To provide this missing result, we have now completed the evaluation by reproducing the Real2Code pipeline ourselves. We have updated Table 3 with the physical executability result from this implementation.
>
> | Method | ALL | ID Classes (Simple) | ID Classes (Complex) | OOD Classes (Simple) | OOD Classes (Complex) |
> | :- | :-: | :-: | :-: | :-: | :-: |
> | Real2Code | 41% | 55% | 37% | 34% | 19% |
>
> ## Q1: Determination of Object Properties at Inference
>
> Thank you for the question. Our model infers both the object category and the number of parts directly from the input point cloud, without requiring annotations.
> **For the object category**: As shown in Section 3.3 and our Appendix, our model's output directly includes semantic part categories (e.g., "link_0": "base_cabinet[SEG]"). This demonstrates that the model implicitly understands the overall object category by deconstructing it into its constituent parts during the generation process. Separately, one could also directly query our ShapeLLM backbone for the object's category, as this is an inherent capability from its pre-training.
> **For the number of parts**: This is not an input but an outcome of the autoregressive process. The model generates the structure joint-by-joint and implicitly learns when to stop, determining the final number of parts by the length of the predicted joint list.
>
> ## Q2: Performance Breakdown and Failure Analysis
>
> Thank you for the question. To provide a more detailed error analysis for physical executability, we have conducted a deeper failure analysis. Due to the rebuttal's length constraints, we selected a representative subset of 3 Out-of-Distribution (OOD) categories (Window, chair, Globe) for this analysis. The results are showed below:
>
> | Category | Failure Rate (%) | Json format Error | Joint Type Error | Joint Axis Error | Joint Origin Error | Subjective Failure Rate (%) |
> | :-: | :-: | :-: | :-: | :-: | :-: | :-: |
> | Window | 15.5 | 0 | 0.03 | 0.05 | 0.07 | 17.2 |
> | chair | 22.2 | 0.01 | 0.06 | 0.06 | 0.09 | 23.5 |
> | Globe | 14.8 | 0 | 0.03 | 0.05 | 0.07 | 18.0 |

---

> > ### Comment · Reviewer_qimd · 2025-08-04
> >
> > Thank you for addressing my questions! I will increase my rating from 4 to 5.

---

> ### Author Response · Authors · 2025-08-05
>
> Thank you for your valuable feedback and for raising our score. Your insights significantly improved our paper's rigor and clarity, and we will incorporate all discussed changes into the revision.

---

### Official Review · Reviewer_KtPh · 2025-06-25

**Clarity:** 2
**Significance:** 3
**Originality:** 3
**Rating:** 4
**Confidence:** 3

**Summary:**

This paper proposes URDF-Anything, a method that, given a single or multiple views of an object, is able of segmenting its parts and predicting the joint parameters of the parts that can be moved (e.g., the hinge of a door). The method is tested on an extensive benchmark and reports metrics on part segmentation and joint prediction as well.

**Questions:**

1. [43-44]: "In addition, we introduce a novel dynamic [SEG] token mechanism [...]". From the paper content, I struggle to see the novelty in this mechanism, as it was introduced by LISA [1] and later adopted by various methods for similar tasks [2,3]. It should be discussed more in detail how the paper modify this token mechanism with respect to prior work.
2. About the physical executability, is there an error anlysis about the sources of error? E.g., percentage of cases in which the URDF is not correctly formed, and percentage in which the format is correct, but the joints are wrong
3. Tab1: "count accuracy" is not defined in the text
4. Tab2: what is "oracle"?
5. Tab2: Why are Articulate-Anything class split not reported?
6. [473-484]: from this example, it seems that the network relies on prior knowledge about the object, i.e. the fact that it has N joints. Is this correct? If it is, this is an important assumption for the method, and it should be clearly stated in the paper.

[1] Lai, Xin, et al. "Lisa: Reasoning segmentation via large language model." CVPR 2024.
[2] Huang, Kuan-Chih, et al. "Reason3d: Searching and reasoning 3d segmentation via large language model." 3DV 2025.
[3] He, Shuting, et al. "Segpoint: Segment any point cloud via large language model." ECCV 2024.

**Ethical Concerns:**

["NO or VERY MINOR ethics concerns only"]

**Final Justification:**

The authors clarified all the main points I raised in my review. After considering also the results provided to other reviewers requests (e.g., the additional ablation on the 3D network architecture, and the context fusion mechanism), I raise my recommendation to Borderline accept. I did not go for a full Accept as I still find the [SEG] token contribution a bit weak considered how it was previously used in the literature. Nonetheless, the paper is solid and the task and results are significant.

**Limitations:**

yes

**Quality:**

2

**Strengths And Weaknesses:**

Strengths

- The task is significant due to its application to real-world scenarios
- The method is simple, and the paper organization explains it very clearly
- The results clearly explain the efficacy of the method, particularly the ones on the physical executability.

Weakness

- The novelty of the paper is unclear, it would be helpful to briefly summarize the contributions at the end of the introduction section. As one contribution the authors state (at 44) “a novel dynamic [SEG] token mechanism”, which is the one used in LISA [1]. Moreover, in 3D this technique has already be used in [2,3], to name a few. The novelty in its use is not clear to me (see question 1).
- The paper is lacking some explainations on the modality used during evaluation (see questions 3-4-5-6)

[1] Lai, Xin, et al. "Lisa: Reasoning segmentation via large language model." CVPR 2024.

[2] Huang, Kuan-Chih, et al. "Reason3d: Searching and reasoning 3d segmentation via large language model." 3DV 2025.

[3] He, Shuting, et al. "Segpoint: Segment any point cloud via large language model." ECCV 2024.

---

> ### Author Rebuttal · Authors · 2025-07-31
>
> We sincerely thank the reviewer for their positive assessment. We thanks for your recognition of our task's real-world significance, the clarity of our method and paper organization, and the efficacy of our results, especially the physical executability metric.
>
> ## W1 & Q1: Novelty & Re-use of the [SEG] Token
>
> Thanks for your valuable feedback.
>
> ## Primary contribution
>
> Our work's primary contribution is not a minor technical improvement, but rather the introduction of a fundamental **Paradigm Shift** to the field of articulated object reconstruction.
> Prior to our work, the prevailing paradigm in this field suffered from several inherent limitations. It typically relied on multi-stage, error-prone pipelines—for instance, using one model to segment parts and another to predict parameters. At the input stage, to simplify the problem, these methods either depended on representations with significant loss of geometric detail (e.g., OBBs) or were limited by 2D images, making it difficult to infer precise 3D structures. On the modeling front, they either employed specialized, small-scale models lacking world knowledge and generalization capabilities, or adopted methods requiring slow, per-instance online optimization (such as those based on Gaussian splatting), which are difficult to scale.
> Our work champions and implements **a new paradigm of end-to-end joint reasoning**. We are the first to demonstrate that a unified 3D Multimodal Large Language Model (3D MLLM) can directly take a **complete 3D point cloud** as input and, through **joint optimization**, produce a functional model that is both **geometrically precise and kinematically consistent** in a single, end-to-end pass. This new paradigm leverages the powerful prior knowledge and reasoning capabilities of large models to fundamentally address the problems of information loss, error accumulation, and poor generalization that plagued traditional methods. The most direct evidence of its effectiveness is that our model achieves a **greater than 50% improvement in Physical Executability**, directly validating the robustness and superiority of this new paradigm.
>
> ## Innovative application of the [SEG] token
>
> The innovation lies in being the first to fundamentally adapt and extend its application and mechanism for end-to-end articulated model reconstruction. This is primarily reflected in the following three core aspects, which also represent the essential differences from prior works like SegPoint and Reason3D:
>
> - **Operating Scale:** SegPoint and Reason3D both operate at the scene level. In contrast, our URDF-Anything focuses on fine-grained part-level analysis.
> - **Function of Segmentation:**  In our framework, segmentation plays another critical role--**A Booster:** Through end-to-end joint training, the precise segmentation task in turn provides strong geometric constraints for the prediction of kinematic parameters, thereby improving their accuracy. As shown in our ablation study (Table 1 below), removing the segmentation task leads to a significant increase in kinematic prediction error, validating its "booster" effect.
> - **Implementation Mechanism:** Both SegPoint and Reason3D directly use the hidden state of the [SEG] token ($h_{seg}$) to guide the segmentation decoder. We have made a key improvement upon this: we fuse the hidden state $h_{seg}$ with the hidden state of its immediately preceding "category token" (e.g., "drawer"), $h_{category}$, resulting in $h_{combined} = [h_{seg} ; h_{category}]$. This design injects strong, part-aware context into the segmentation process, which is crucial for distinguishing between multiple visually similar parts on the same object. We validated this through a dedicated ablation study (Table 2 below), which shows our context fusion design achieves superior segmentation accuracy.
>
>
> To experimentally support the above claims, we conducted two key ablation studies:
>
> Table 1: Validating the "Booster" Effect of Segmentation on Kinematic Prediction
> | Method | Type Error ↓ | Axis Error ↓ | Origin Error ↓ |
> | :-: | :-: | :-: | :-: |
> | URDF-Anything w/ seg (Full Model) | 0.008 | 0.132 | 0.164 |
> | URDF-Anything w/o seg (No Seg) | 0.009 | 0.138 | 0.175 |
>
> Table 2: Validating the Superiority of the Context Fusion Mechanism
> | Segmentation Mechanism | mIoU ↑ |
> | :-: | :-: |
> | Context Fusion (Ours) | 0.63 |
> | Generic $h_{seg}$ only | 0.58 |
>
> Finally, we will revise the end of our introduction to clearly summarize our core contributions, including: (1) Proposing the first end-to-end 3D MLLM framework for articulated object reconstruction, championing a new paradigm from complete 3D input to joint prediction output; (2) Achieving deep coupling and joint prediction of kinematic parameters and geometric segmentation through an innovative application of the [SEG] token; (3) Demonstrating the superiority of this new paradigm through extensive experiments. And we add citations to these relevant works [1, 2, 3] in our revised paper.
>
>
>
> ## W2: Explanation on Input Modality
>
> We thank the reviewer for pointing out that the explanations regarding the input modalities for evaluation could be clearer. We agree and appreciate the opportunity to clarify this here and in revision.
>
> As the reviewer noted, we have provided more details in our answers to the specific Q 3, 4, 5, and 6.
>
> ## Q2: Error Analysis on Physical Executability
>
> Thanks for this  question. As your suggestion, we have conducted a detailed analysis of the failure cases for our method. The results are as follows, break down the 22% overall failure rate observed in Table 3.
>
> Table 3: Breakdown of failure cases for URDF-Anything on the physical executability task.
> | Overall Failure| JSON Format Error | Incorrect Joint Parameters |
> |:-:|:-:|:-:|
> | 22% | 1% | 21% |
>
> This analysis reveals that the vast majority of failures (**21%** out of 22%) are due to the prediction of incorrect or physically implausible joint parameters (e.g., origin, axis). In contrast, structural errors, such as generating a malformed JSON or an invalid URDF file, are extremely rare (**1%**).
>
> This breakdown confirms that our MLLM-based framework is highly proficient at learning the correct syntax and structure for URDF generation. The primary remaining challenge lies in the precise quantitative prediction of kinematic parameters, not in generating a valid file format.
>
> Due to space constraints, we did not include this analysis in the initial submission. We will add this table and the corresponding discussion to the experimental section in the revision to provide a more comprehensive evaluation.
>
> ## Q3: "Count Accuracy" Metric
>
> Thank you for pointing out this omission. You are correct, we should have defined this metric in the text. **"Count Accuracy"** measures the percentage of test samples for which the number of predicted articulated parts (i.e., links with non-fixed joints) exactly matches the ground-truth number of parts.
> We will add this definition to Section 4 (Evaluation Metrics) in the final version of the paper.
>
> ## Q4: "Oracle" Term
>
> Thank you for the question.
> The "Oracle" setting in Table 2, a protocol from the Articulate-Anything paper, provides baselines with ground-truth part segmentations. This is to fairly evaluate their kinematic prediction module in isolation, by removing errors from their preceding segmentation step.
>
>
> ## Q5: Baseline Results Split
>
> Thank you for pointing out this omission.
> To clarify our experimental setup, our data split is consistent across the baselines. Our In-Distribution (ID) split consists of the five categories shown in our supplementary materials: "StorageFurniture", "Laptop", "Box", "Table", and "Refrigerator". The remaining 41 object categories from the PartNet-Mobility dataset constitute our Out-of-Distribution (OOD) split.
>
> ## Q6: Concern about Reliance on Prior Knowledge
>
> Thank you for the question. The example you saw in the appendix is indeed one of the diverse templates we used during training, but it does not mean our method must rely on this prior knowledge at inference time. We apologize for any potential misunderstanding the appendix may have caused and wish to offer a detailed clarification here.
> Our approach is divided into two phases: training and inference.
>
> **During Training**: We designed highly diverse prompt templates to enhance the model's robustness. A portion of these templates did include the "number of parts," primarily serving as a pedagogical tool or a strong supervisory signal to ease the fitting pressure and help the model learn the correlation between an object's visual form and its internal articulated structure more effectively. However, our training data also contained numerous generic templates that did not provide this information, which compelled the model to learn to infer the number of parts autonomously from visual and geometric features, rather than relying solely on the prompt.
>
> **During Inference (Testing)**: Providing the number of parts is entirely optional at inference time. It is crucial to emphasize that for all quantitative evaluations in the paper and for fair comparison with other models, we uniformly used a generic prompt that did not contain any information about the number of parts or joints (e.g., "Please segment the object and predict its joint parameters").
>
> We will revise the appendix to feature a simpler object JSON as an example for better clarity.

---

> > ### Comment · Reviewer_KtPh · 2025-08-04
> > **Concerns mostly addressed - Results could be provided for Q5**
> >
> > I thank the authors for clarifying some terminology and for the better explaination on the use of the SEG token. I still believe that the missing results in Tab2 should be provided for completeness. As similar results were reported in Tab3, I think the metadata is available to report the separate results on the two splits.

---

> ### Author Response · Authors · 2025-08-05
> **Response with Requested Results for Table 2**
>
> We appreciate your feedback and are pleased that our clarifications on the terminology and the [SEG] token were helpful.
>
> Regarding your question about the table data for Articulate-Anything, I would like to offer the following explanation:
>
> We initially reported only a single "ALL Classes" score because Articulate-Anything is a **training-free, retrieval-based method**. Since it is **not trained on our In-Distribution (ID) dataset**, the ID vs. OOD distinction is not directly relevant to it in the same way it is for trained models. This reporting style is also consistent with the evaluation in the original Articulate-Anything paper.
>
> To provide the most complete comparison and fully address your request for a split analysis, we have re-evaluated Articulate-Anything on our specific ID and OOD splits.The complete results are presented in the table below:
>
> | Method | All Classes Type ↓ | All Classes Axis ↓ | All Classes  Origin ↓ | ID Classes Type ↓ | ID Classes Axis ↓ | ID Classes  Origin ↓ | OOD Classes Type ↓ | OOD Classes Axis ↓ | OOD Classes Origin ↓ |
> | :--- | :---: | :---: | :---: | :---: | :---: | :---: | :---: | :---: | :---: |
> | Articulate-Anything | 0.025±0.005 | 0.145±0.450 | 0.207±0.392 | 0.018±0.004 | 0.143±0.198 | 0.195±0.237 | 0.026±0.005 | 0.145±0.480 | 0.208±0.411 |
>
> We will replace the original table with this updated version in the revision.

---

> > ### Comment · Reviewer_KtPh · 2025-08-06
> >
> > Thanks for providing the complete results. Indeed there is very little difference between IID and OOD classes performance, as was expected given the setting.
> > Thanks again to the authors, I do not have further comments for now

---

> > > ### Author Response · Authors · 2025-08-06
> > > **Response to Reviewer KtPh**
> > >
> > > Dear Reviewer KtPh,
> > >
> > > Thank you for your insightful and constructive comments. Your thoughtful feedback has been instrumental in improving our work.
> > >
> > > We have carefully considered all your points and have revised the manuscript accordingly. Below is a point-by-point response to the concerns you raised:
> > >
> > > (1)To resolve your primary concern regarding novelty, we provided a detailed explanation of our "**paradigm shift**" contribution and supported it with **two new ablation studies** that empirically validate our innovative application of the [SEG] token and its "booster" effect. (2)Following your suggestions, we added the requested **error analysis for physical executability** and have **clarified all terminology** you pointed out, such as "Count Accuracy" and "Oracle." (3)Finally, to address your last concern, we have run a **new evaluation** to provide the complete, split results for Articulate-Anything as you requested.
> > >
> > >
> > > Therefore, we hope these responses have adequately addressed your comments. We welcome any further questions you may have. With these improvements, we hope you might reconsider your evaluation of our work, as your support would be a great encouragement to us.

---

> > > ### Author Response · Authors · 2025-08-07
> > > **Following up on our discussion**
> > >
> > > Dear Reviewer KtPh,
> > >
> > > Thank you again for your detailed feedback and constructive engagement, which has significantly improved our manuscript.
> > >
> > > We have submitted comprehensive responses addressing all the points you raised, supported by new experiments and manuscript revisions.
> > >
> > > We hope our responses and revisions have fully addressed your concerns. We would be very grateful if you would consider updating your evaluation. Please let us know if any points need further clarification.
> > >
> > > Best regards,
> > > The Authors

---

### Official Review · Reviewer_euKz · 2025-07-01

**Clarity:** 3
**Significance:** 3
**Originality:** 3
**Rating:** 5
**Confidence:** 5

**Summary:**

This paper presents URDF-Anything, a novel end-to-end framework that leverages 3D Multimodal Large Language Models (3D MLLMs) to automatically reconstruct articulated object digital twins in URDF format from visual observations (images or point clouds). The method introduces a specialized [SEG] token mechanism to jointly predict object segmentation and kinematic parameters. Experiments on the PartNet-Mobility dataset show substantial improvements in segmentation accuracy, kinematic prediction precision, and physical executability over prior baselines.

**Questions:**

See weaknesses.

**Ethical Concerns:**

["NO or VERY MINOR ethics concerns only"]

**Final Justification:**

My concerns have been well addressed during the rebuttal. As a compositional work, my primary concern was the novelty. The authors have convincingly demonstrated that their framework is technically non-trivial during the rebuttal. The integration of components is the result of careful design, and, to the best of my knowledge, their framework is the first to introduce the point cloud modality for LLM-based URDF generation. While additional ablation studies could further strengthen the work, I believe the proposed framework represents a meaningful contribution to this less-explored field.

I recommend accepting the paper, as the authors have committed to releasing the model and code, which will be highly beneficial for the community.

**Limitations:**

yes

**Quality:**

2

**Strengths And Weaknesses:**

Strengths:
+ The paper addresses an important and underexplored problem in robotic simulation: automatic articulated object modeling and URDF file generation.

+ The framework is end-to-end, combining geometry understanding and kinematic reasoning.

+ Clear reproducibility, and good experimental results compared to former works.


Weaknesses:
- While the use of the [SEG] token for reasoning-based segmentation is intuitive and well-motivated in the LISA framework, its extension in this paper raises certain concerns. In the context of this work, applying [SEG] to guide 3D point cloud segmentation is understandable, since the token can serve as a semantic anchor to align text-based part labels with geometric features. However, what is less clear—and somewhat black-box in nature—is how the model is able to simultaneously infer accurate joints using the same mechanism. The paper lacks a concrete explanation or ablation showing how the [SEG] token contributes to learning the kinematic structure, as opposed to just part segmentation. It almost appears as if the model “magically” learns to associate geometry with articulation semantics, without clearly defined inductive biases or intermediate supervision. A deeper analysis or visualization would greatly help demystify this process.

- This paper lacks sufficient ablation studies to support its methodological choices. Most components are fixed without justification or comparison. In particular, the choice of Uni3D as the 3D backbone is taken for granted, but alternative backbones should have been evaluated for comparison. A more thorough experimental analysis, similar to what was done in the paper Articulate-Anything, would significantly strengthen the paper. That work serves as a good reference for the types of diagnostic or ablation studies that could be conducted. While I will not list them all here, following a similar level of evaluation would be beneficial for this paper.

- My key concern is that this is a compositional work that primarily integrates existing ideas rather than introducing novel concepts. Each component of the paper—whether it's the use of 3D multimodal large language models (3D MLLMs), LoRA for fine-tuning, the incorporation of [SEG] tokens, point cloud generation, mesh reconstruction, or structured output formatting—is based on prior methods. While the system as a whole may function effectively, there is limited novelty in the individual technical contributions. The framework appears to repackage existing tools and techniques into a cohesive pipeline rather than pushing the boundaries of any specific subfield. Please let me know if the authors have any arguments regarding the novelty of this work.

---

> ### Author Rebuttal · Authors · 2025-07-31
>
> We sincerely thank the reviewer for their positive feedback. We are particularly encouraged by your recognition of the importance of the problem we are addressing, the integrated nature of our end-to-end framework, and our strong experimental results and clear reproducibility.
>
> ## W1: Unclear Mechanism of the [SEG] Token for Kinematics
> Thank you for this insightful and critical question. In our paper (lines 47-50), we described this as a "tight coupling" that "ensures full consistency between predicted kinematics and reconstructed geometry". We would like to take this opportunity to "demystify" this mechanism with both a theoretical explanation and new experimental validation.
>
> **1. Theoretical Explanation:  Geometric Regularization via Joint Optimization**
> Our model does not learn this association "magically"; the core principle is Geometric Regularization, an effect that emerges from our end-to-end joint optimization.
> Essentially, the segmentation task ($L_{seg}$) provides a powerful physical anchor for the kinematic prediction task ($L_{text}$). A model predicting only kinematics might hallucinate a joint that is textually plausible but physically baseless (e.g., a rotation axis outside the object). Our joint segmentation task forces the model to ground its abstract kinematic predictions (e.g., a rotation axis) in the concrete geometric entities of the point cloud. To accurately segment a door, the model must implicitly learn where its center of rotation and axis should be. This geometric grounding effectively constrains the parameter search space, leading to more physically consistent and accurate articulation structures.
>
> This is enabled by the Shared Representation in our end-to-end model. Because both tasks share most network parameters and are optimized via a unified loss function ($L = λ_{text}*L_{text} + λ_{seg}*L_{seg}$), the model must learn an internal representation effective for both. This means the hidden states used to generate kinematic parameters must also contain information useful for the subsequent geometric segmentation, thus tightly coupling them in the latent space.
>
> **2. Experimental Validation: From Ablation to Visualization**
> To empirically validate this theory, we will add two key experiments to our revised manuscript to demonstrate how the [SEG] token facilitates kinematic learning:
> - **Ablation Study: Quantifying the Contribution of Segmentation.**
>     - **Design:** We will train a baseline model where the [SEG] token and its associated segmentation loss ($L_{seg}$) are removed. This model will predict kinematic parameters using only the language modeling loss ($L_{text}$). We then compare its parameter accuracy against our full model.
>     - **Results:** As shown in Table 1,  without the geometric regularization from the segmentation task, this baseline has higher error in its kinematic predictions. This result quantitatively proves that the segmentation task is crucial for learning accurate kinematic structures.
>
> Table 1: Quantifying the Contribution of Segmentation.
> | Method | Type Error | Axis Error | Origin Error |
> | :-: | :-: | :-: | :-: |
> | URDF-Anything w/ seg | 0.008 | 0.132 | 0.164 |
> | URDF-Anything w/o seg | 0.009 | 0.138 | 0.175 |
>
> - **Attention Visualization: Opening the "Black Box".**
>     - **Design:** To visually inspect how the model grounds its kinematic predictions, we visualized the self-attention distribution from a specific layer of the two model versions in our Ablation Study: our full model (URDF-Anything w/ seg) and the baseline without the segmentation task (URDF-Anything w/o seg). This allowed us to observe which parts of the input point cloud the models focus on when generating a kinematic parameter token (e.g., 'axis').
>     - **Results:** The visualizations (which can't be shown here and we will add to the revision) show that when generating kinematic tokens, our full model's attention is highly concentrated on the physically relevant joint regions of the point cloud (e.g., hinges, sliders). In contrast, the attention of the model trained without the segmentation loss is far more diffuse and unfocused.
>     - **Significance:** This visualization provides direct, intuitive evidence at a "neural level" of how our joint training forces the model to associate abstract kinematic concepts with concrete geometric features. It demystifies the process, showing that the connection is not "magic" but a learned behavior enforced by the geometric regularization of the segmentation task, directly addressing your concern about the "black-box" nature of the mechanism.
>
> We will incorporate these new experiments, including the visualizations, into our revised manuscript. Thank you again for pushing us to clarify this crucial aspect of our work.
>
>
>
> ## W2: Insufficient Ablation Studies
>
> We sincerely thank you for your valuable suggestion regarding the ablation studies.
>
> We would like to take this opportunity to clarify the core focus of our current work. Our primary objective is to be the first to demonstrate and validate the feasibility and superiority of a novel, unified 3D MLLM framework for the task of articulated object reconstruction. Through this, we aim to help guide the field's evolution from traditional multi-stage or optimization-based paradigms toward a more efficient and versatile **end-to-end large model paradigm.** Based on this core objective, our experimental design (including the ablation study in Section 4.4) prioritized validating the most fundamental hypothesis of this new paradigm: that **multimodal inputs combining explicit 3D point clouds and text instructions** are crucial compared to relying solely on 2D images or pure geometric information. We believe that in the initial stages of establishing a completely new framework, proving the correctness of its core design philosophy is more critical than optimizing one of its replaceable components.
>
> However, we also agree that conducting more comprehensive diagnostic experiments would better demonstrate the superiority and robustness of our unified MLLM joint prediction architecture. Therefore, we have supplemented our work with a new preliminary experiment. Specifically, we replaced the original Uni3D backbone with a classic alternative, PointNet++, while keeping all other parts of the model (such as the 3D MLLM, the [SEG] token mechanism, and the training strategy) unchanged.
>
> Table 2: Ablation Study on 3D Backbone
>
> | Method | mIoU ALL ↑ | mIoU ID ↑ | mIoU OOD ↑ | Count Acc ALL ↑ | Count Acc ID ↑ | Count Acc OOD ↑ |
> | :-: | :-: | :-: | :-: | :-: | :-: | :-: |
> | **ours (uni3d)** | 0.63 | 0.69 | 0.62 | 0.97 | 0.99 | 0.96 |
> | **ours (pointnet++)** | 0.57 | 0.66 | 0.56 | 0.96 | 0.99 | 0.95 |
>
> Our preliminary results show the following:
>
> **Validation of Our Choice:** When using PointNet++ as the backbone, the model's overall performance sees a slight decrease compared to using Uni3D. This, to some extent, validates our initial choice of Uni3D as a powerful and suitable feature extractor.
>
> **Demonstration of Framework Robustness:** Despite the slight performance drop, this new PointNet++-based variant still significantly outperforms all baseline methods compared in our paper across all key metrics.
>
> This strongly demonstrates that the success of our framework is not merely dependent on a single specific SOTA component; its core advantage stems from the effectiveness of our proposed unified MLLM joint prediction architecture. This result also reaffirms that there is ample room for future research to conduct more detailed component-level comparisons and optimizations (e.g., exploring more efficient backbones) upon our framework to further enhance performance.
>
>
>
> ## W3: Limited Novelty / Compositional Work
>
> We appreciate the opportunity to clarify our perspective on the novelty of URDF-Anything.
>
> We respectfully argue that the novelty of our work lies not in inventing a single, isolated algorithm, but in successfully conceiving, implementing, and validating a **new end-to-end paradigm** for articulated object reconstruction. Within this new paradigm, we performed crucial, non-trivial adaptations of existing technologies to solve challenges unique to this domain.
>
> Previously, the field relied on fragmented, multi-stage pipelines with simplified inputs, which led to error accumulation and physically inconsistent models. Our unified paradigm solves this by processing complete 3D point clouds to **jointly optimize geometry and kinematics in a single pass.** The success of this fundamental shift is demonstrated by a >50% improvement in physical executability and superior accuracy over piecemeal approaches (Table 2 in paper), validating its superior robustness.
>
> This paradigm is not a simple composition; it required key technical innovations and has pushed the boundaries of the subfield:
>
> **Technical Novelty:** We didn't just reuse the [SEG] token; we fundamentally adapted it. Our context-fusion mechanism (merging category and [SEG] token information) provides the essential part-level semantic understanding required for this complex, single-object deconstruction task.
>
> **Pushing Boundaries:** Our most significant contribution is a leap in generalization. By leveraging a 3D MLLM, we shift the field from dataset-specific models to those that can generalize to entirely new object categories. This is proven by our strong performance on out-of-distribution (OOD) objects (Tables 1-3 in paper), a crucial advancement for the field.
>
> In essence, our work provides the first robust "existence proof" for applying 3D MLLMs to this domain. We believe establishing this powerful and effective new baseline is a significant contribution that opens new avenues for future research.

---

> > ### Comment · Reviewer_euKz · 2025-08-06
> >
> > Thanks for the response from the authors, my concerns have been addressed better than I expected. I would appreciate it if the authors could further elaborate on the following two points:
> >
> > 1. In the additional ablation study provided in A1, while there is a reported performance drop without segmentation, the gap appears relatively small. Could the authors clarify how statistically significant these differences are?
> >
> > 2. Regarding the novelty, the current claim still comes across as a well-engineered combination of existing techniques. Could the authors point to any specific component or interaction that fails when naively assembled from prior work (or from your own earlier explorations)? This would help clarify what aspects of the integration are truly novel or technically non-trivial.

---

> > > ### Author Response · Authors · 2025-08-08
> > > **Response to Reviewer euKz**
> > >
> > > ## Q1: Clarifying the Performance Gap
> > >
> > > We sincerely thank the reviewer for this question. The significance of this gap is two fold: 1) in the context of robotics, these small numerical differences have major practical consequences, and 2) a deeper analysis reveals this average gap masks a critical improvement in robustness on the most complex object categories.
> > >
> > > ### 1. The Practical Significance: Why Small Errors Matter in Robotics
> > >
> > > In robotics and physical simulation, precision is paramount. Unlike typical computer vision tasks, small numerical errors in kinematics can lead to complete functional failure.
> > >
> > > **A Concrete Example:** Consider a simple cabinet. An origin error of just **1 cm (0.01m)** can cause a drawer to bind against its frame instead of sliding smoothly. Similarly, an axis error of a mere **3 degrees (~0.05 radians)** can make a hinged door collide with an adjacent wall or cabinet instead of swinging freely in its intended arc.
> > >
> > > These seemingly "small" errors are a primary reason why generated URDF models fail in simulation. While the average error reduction reported in our ablation study (Table 1: Quantifying the Contribution of Segmentation.) is modest, it reflects a significant decrease in these critical, task-failing prediction errors. This directly contributes to the substantial (>50%) improvement in overall Physical Executability we reported in the main paper, which is arguably the most crucial metric for this task.
> > >
> > >
> > > ### 2. Deeper Analysis: Uncovering the Source of Performance Degradation
> > >
> > > To further investigate, we performed a fine-grained analysis of the error distribution. We found that the performance degradation of the w/o seg model is not uniformly distributed across all categories. Instead, it is disproportionately concentrated in a few specific types of challenging objects, particularly those with:
> > >
> > > - Multiple, closely-spaced, and kinematically similar joints (e.g., a cabinet with several thin, stacked drawers).
> > > - Small, intricate moving parts where the geometry of the joint is subtle (e.g., a complex faucet handle or a multi-part switch).
> > >
> > > To understand the underlying reason, we visualized the model's attention distribution during the generation of kinematic parameters. The results were revealing:
> > > - **For the model without segmentation (w/o seg):** When faced with these complex objects, its attention is often diffuse and unfocused. For instance, when trying to predict the axis for "drawer_2", its attention might spread across all three drawers, or incorrectly latch onto a visually salient but kinematically irrelevant feature like a handle. Without the strong geometric grounding provided by the segmentation task, the model gets "confused" by the visual similarity and struggles to isolate the correct geometric entity for its kinematic prediction.
> > > - **For our full model with segmentation (w/ seg):** Its attention is remarkably precise. When predicting the parameters for "drawer_2", the model's attention correctly and sharply concentrates on the geometry of that specific drawer. The joint optimization process effectively teaches the model an essential principle: "To predict the kinematics for this part, I must first accurately attend to its specific geometry, as I will be required to segment it immediately after."

---

> > > ### Author Response · Authors · 2025-08-08
> > > **Response to Reviewer euKz**
> > >
> > > ## Q2: Novelty Beyond Combination
> > >
> > > We sincerely thank the reviewer for this insightful question. It gives us the opportunity to provide a unified perspective on the novelty of our work. Our framework is not a simple combination of existing techniques, but rather **a new paradigm, systematically engineered to overcome the inherent failure modes of a series of "naive" approaches.**
> > >
> > > Our core argument is this: **For the complex task of articulated object reconstruction, any seemingly "direct" or "reasonable" approach of simply combining existing tools will fail at a critical stage.** Our novelty lies precisely in identifying these failure modes and proposing targeted, non-trivial solutions. Below, we will follow a clear logical chain to demonstrate these challenges and our countermeasures one by one.
> > >
> > > ### Step 1: The Foundational Dilemma – Why a Naive "See and Tell" Fails
> > >
> > > At the very start, the most "naive" idea is to use a powerful Vision-Language Model (VLM) to generate a URDF by simply "looking at a picture."
> > >
> > > - **Naive Approach:** Use a state-of-the-art VLM (like Qwen-VL), input a 2D image, and directly predict 3D kinematic parameters.
> > >
> > > - **Failure Mode:** 2D images suffer from inherent ambiguity and a lack of 3D information. A VLM can identify "this is a drawer," but it cannot accurately infer its 3D depth, dimensions, and motion axis from pixels alone, leading to errors so large they are useless for physical simulation (as shown below, errors are often an order of magnitude larger).
> > >
> > > - **Our Non-trivial Solution & Conclusion:** We determined that **using a complete 3D point cloud as input is a non-negotiable prerequisite** for achieving physics-level precision. This foundational choice fundamentally distinguishes our work from all 2D-based "see and tell" approaches.
> > >
> > > **Table 1: Failure of Image-based Input for Precise Parameter Prediction** (adapted from Table 4 in the main paper)
> > > | Method Variant | Input Modality | Type Error ↓ | Axis Error ↓ | Origin Error ↓ |
> > > | :--- | :--- | :---: | :---: | :---: |
> > > | Qwen-VL (Image-based) | Image + Text | 0.38 | 0.81 | 0.18 |
> > > | Our Method | Point Cloud + Text | 0.008 | 0.132 | 0.164 |
> > >
> > >
> > >
> > > ### Step 2: The Processing Pipeline Dilemma – Why a Naive "Divide and Conquer" Fails
> > >
> > > Having established the necessity of 3D input, the next "reasonable" engineering idea is to adopt a "divide and conquer" or decoupled pipeline.
> > >
> > > - **Naive Approach:** Build a decoupled system where part segmentation and kinematic parameter prediction are handled by two separate, independently trained models.
> > > - **Failure Mode:** This leads to error propagation and contextual disconnection. The segmentation model is blind to kinematics, and the kinematics model cannot leverage geometry to refine its predictions. For example, the model cannot use the knowledge that "a door must rotate around a hinge" to, in turn, optimize the segmentation of the "door" and "hinge."
> > > - **Our Non-trivial Solution & Conclusion:** We introduce an **end-to-end joint reasoning paradigm.** Within this framework, geometric segmentation and kinematic prediction are optimized simultaneously, forming **a joint probability distribution.** Segmentation is no longer an isolated upstream step but acts as a powerful **"geometric booster"**, providing strong physical constraints for kinematic prediction. As the table below shows, removing the segmentation task leads to a degradation in all kinematic metrics, directly proving the necessity and superiority of joint optimization.
> > >
> > > **Table 2: Validating the "Booster" Effect of Segmentation on Kinematic Prediction** (Adapted from our response to W1 of Reviewer euKz for convenience)
> > > | Method | Type Error ↓ | Axis Error ↓ | Origin Error ↓ |
> > > | :--- | :---: | :---: | :---: |
> > > | URDF-Anything w/ seg (Full Model) | 0.008 | 0.132 | 0.164 |
> > > | URDF-Anything w/o seg (No Seg) | 0.009 | 0.138 | 0.175 |

---

> > > ### Author Response · Authors · 2025-08-08
> > > **Response to Reviewer euKz**
> > >
> > > ### Final Argument: Success Stems from the Architecture, Not a Single Component
> > >
> > > Finally, to address the concern that our success might just be an artifact of using a particularly powerful component (like Uni3D), we demonstrate our framework's architectural robustness.
> > >
> > > - **Naive Assumption:** The framework's success is merely a "dividend" from a single state-of-the-art component.
> > >
> > > - **Our Conclusion & Evidence:** We swapped the Uni3D backbone with the classic PointNet++. While performance slightly decreased, the new combination still comprehensively and significantly outperformed all baseline methods in the paper. This proves that our success stems from the fundamental superiority and robustness of the proposed framework itself, not from a dependency on any single component.
> > >
> > > **Table 5: Validating Architectural Robustness with Different 3D Backbones** (Adapted from our response to W2 of Reviewer euKz for convenience)
> > > | Method | mIoU ALL ↑ | mIoU ID ↑ | mIoU OOD ↑ | Count Acc ALL ↑ | Count Acc ID ↑ | Count Acc OOD ↑ |
> > > | :-: | :-: | :-: | :-: | :-: | :-: | :-: |
> > > | **ours (uni3d)** | 0.63 | 0.69 | 0.62 | 0.97 | 0.99 | 0.96 |
> > > | **ours (pointnet++)** | 0.57 | 0.66 | 0.56 | 0.96 | 0.99 | 0.95 |
> > >
> > > ### In Summary
> > >
> > > In conclusion, our work is not a "well-engineered combination" but **a systematic, bottom-up process of innovation**. Starting from the most fundamental choice of input modality, we identified and solved the failure modes of "naive" approaches in processing pipelines, reasoning mechanisms, and core signaling. Each decision was driven by a deep understanding of the preceding challenge and is supported by robust ablation studies. The final result is a new paradigm that is logically coherent, architecturally robust, and achieves a breakthrough in performance.

---

> ### Author Response · Authors · 2025-08-06
> **Response to Reviewer euKz**
>
> Dear Reviewer euKz,
>
> We would like to extend our sincere thanks for your thorough review and valuable feedback on our paper.
>
> To address the points raised in your review, we have performed additional experiments and revised the paper. Here is a summary of the main changes:
>
> (1)On the "black-box" mechanism: We added a **theoretical explanation** and **two new experiments** (an ablation and a visualization) to demystify how our model learns kinematics. (2)On insufficient ablations: We conducted **a new ablation study** on the 3D backbone (using PointNet++) to justify our architectural choice and prove our framework's robustness. (3)On novelty: We clarified that our core contribution is establishing a **new, end-to-end paradigm**, whose superiority is proven by major gains in physical executability and generalization.
>
> We have done our best to address all your suggestions and hope you find our revisions satisfactory. We are happy to engage in further discussion if needed. Should our response resolve your concerns, we would be deeply appreciative if you would consider raising your score.

---

> ### Author Response · Authors · 2025-08-08
> **Response to Reviewer euKz**
>
> ## Q2: Novelty Beyond Combination
>
> ### Step 3: The Interaction Mechanism Dilemma –  Why a Naive Static segmentation Fails
> Within our joint framework, a crucial question remains: how should the LLM interact with the 3D features to produce a segmentation? This section explains why our dynamic [SEG] mechanism is superior to simpler, static alternatives.
>
> - **Naive Approach:** A naive but plausible approach is to treat segmentation as a static, post-hoc classification task. This is represented by our Uni3D w/ text baseline, where a model tries to match pre-defined text labels (e.g., "drawer") to geometric regions in the point cloud, decoupled from the main kinematic reasoning process.
> - **Failure Mode:** This approach fails because the segmentation signal (a static text label) is **context-poor**. It is ignorant of the object's overall kinematic structure (e.g., which drawer this is in a stack) and the specific motion parameters being predicted. This decoupling prevents the model from learning the crucial correlation between a part's geometry and its function.
>
> - **Our Non-trivial Solution & Conclusion:** Our superiority stems from two fundamental differences. **First**, our [SEG] token is not a static label but a **dynamic, context-rich signal**. It is generated by the LLM after it has already processed both the complete point cloud geometry and the preceding kinematic parameters (e.g., parent-child links). Its embedding is therefore inherently informed by both geometric and structural context. **Second**, our framework learns a **joint probability distribution over both structure and geometry**. This transforms segmentation from a simple matching task into a context-aware grounding problem.
>
> - **Evidence:** Our ablations prove these two points.
>     - To prove the necessity of **geometric context**, the **"Geometry-Blind"** experiment denies the model access to the point cloud. The signal becomes ungrounded, and the model fails completely (mIoU ≈ 0.1).
>     - To prove the necessity of **structural (kinematic) context**, the **"Segmentation-Only"** experiment removes the kinematic prediction task. This model (mIoU 0.61) is consistently outperformed by our full model (mIoU 0.63), showing that kinematic context refines segmentation.
>
>
> **Table 3: Ablation on Key Components for Segmentation** (Adapted from our response to W3.3 of Reviewer qimd for convenience)
> | Model | Segmentation Accuracy (mIoU ↑) | Count Accuracy (Count Acc ↑) |
> | :-: | :-: | :-: |
> | Uni3D w/ text | 0.54 | 0.84 |
> | Our Full Model | 0.63 | 0.97 |
> | | | |
> | Ablation 1: Segmentation-Only | 0.61 | 0.96 |
> | Ablation 2: Geometry-Blind | ~0.1 (Near-random) | - |
>
>
> ### Step 4: The Core Mechanism Dilemma – Why a Generic Segmentation Signal Fails
>
> Even with the right framework, the challenge of generating precise segmentation commands for multiple similar parts on a single object remains.
>
> - **Naive Approach:** A naive approach would be to directly apply the generic [SEG] token mechanism, inspired by its success in 2D image segmentation. In that paradigm, the token typically acts as a universal pointer to an object or region.
>
> - **Failure Mode:** This generic approach is insufficient for our task. A universal [SEG] token lacks the fine-grained context needed to distinguish "drawer 1" from "drawer 2" and will fail when faced with the ambiguity of single-object, multi-part reconstruction.
>
> - **Our Non-trivial Solution & Conclusion:** We designed a task-specific "Context Fusion" mechanism that merges the [SEG] token's state with that of its preceding category token. This seemingly small modification is a key innovation for resolving part-level ambiguity. Experiments confirm that this targeted design significantly outperforms the generic application.
>
> **Table 4: Validating the Superiority of the Context Fusion Mechanism** (Adapted from our response to W1 of Reviewer KtPh for convenience)
> | Segmentation Mechanism | mIoU ↑ |
> | :--- | :---: |
> | **Context Fusion (Ours)** | **0.63** |
> | Generic $h_{seg}$ only | 0.58 |

---

> ### Comment · Reviewer_euKz · 2025-08-08
>
> Thanks for the authors’ reply. I believe that steps 3 and 4 in A2 have largely addressed my concern regarding novelty. I would encourage adding a statement in the paper to explicitly note that this compositional work (i.e., integrating different modules) is technically non-trivial.
>
> I have one additional short question: in the specific field of URDF generation, when leveraging an MLLM, are you the first to introduce the point cloud modality? If so, this could also be highlighted as a novelty point. If not, could you please provide relevant references and explain why your approach is superior?

---

> > ### Author Response · Authors · 2025-08-08
> > **Response to Reviewer euKz**
> >
> > Dear Reviewer,
> >
> > We are very grateful for your positive feedback and are delighted to know that our previous response has largely addressed your concerns regarding the novelty of our work. This is of great encouragement to us.
> >
> > We fully agree with and will adopt your valuable suggestion. In the revision of the paper, we will add a statement to explicitly emphasize that our systematic integration is technically non-trivial.
> >
> > Regarding the final question you raised, our answer is:
> >
> > **Yes, to the best of our knowledge, in the specific field of URDF generation using MLLMs, we are indeed the first to introduce and utilize the 3D point cloud modality as the core input for the framework.**
> >
> > We believe this is a crucial design choice, and the detailed reasoning behind it is elaborated in **Section 4.4, "Ablation Study,"** of our paper. Furthermore, following your insightful feedback, we will also explicitly highlight that "being the first to introduce the point cloud modality" is one of the core novelty points of our work in the revision to make it more prominent.
> >
> > Once again, we sincerely thank you for your valuable time, insightful feedback, and constructive suggestions throughout the review process. Your guidance has been invaluable in helping us to improve and polish our paper. We hope that these clarifications have fully addressed your final concerns and would be grateful if you would reconsider the **score** for our paper.

---

> > > ### Comment · Reviewer_euKz · 2025-08-08
> > >
> > > Thanks to the authors for their reply. My concerns have been well addressed during the rebuttal, and I have no further questions. While additional ablation studies could have strengthened the work, I believe the proposed framework remains a meaningful contribution to this less-explored field for the community.
> > >
> > > I am raising my rating to Accept. I hope the authors will keep their promise to add the attention visualizations to the main paper and include the new ablation studies in the appendix or supplementary material.
> > >
> > > By the way, kudos to the authors who actively participated in the rebuttal. The other co-authors should treat them to a nice meal for such a thorough and thoughtful response. Wishing you all a well-deserved rest after the rebuttal period.

---

> > > > ### Author Response · Authors · 2025-08-08
> > > >
> > > > Thank you very much for your reply and positive feedback. We are delighted to hear that our rebuttal has successfully addressed your concerns.
> > > >
> > > > We sincerely appreciate you raising your rating to "Accept". Your recognition of our work is a great encouragement to us. We would like to confirm that we will keep our promise to add the attention visualizations to the main paper and include the new ablation studies in the appendix/supplementary material in the final version.
> > > >
> > > > Thank you again for your valuable time, constructive comments, and support for our work. We also appreciate your kind and encouraging words at the end—we will certainly pass on your suggestion to the co-authors!

---

> > > > > ### Comment · Reviewer_euKz · 2025-08-08
> > > > >
> > > > > Thanks. One last question: do you plan to release your model and code? This would be highly beneficial for the community.

---

> > > > > > ### Author Response · Authors · 2025-08-08
> > > > > >
> > > > > > Yes, absolutely. We plan to make the code and trained models publicly available to support reproducibility and future work, upon the paper's acceptance. Thank you for the question.

---

### Official Review · Reviewer_xYGo · 2025-07-08

**Clarity:** 3
**Significance:** 3
**Originality:** 3
**Rating:** 5
**Confidence:** 4

**Summary:**

The paper proposes an Multi-modal LLM based pipeline for estimating the URDF of articulated objects. The input is a single or multi-view RGB images from which pointcloud is extracted using off the shelf methods like Dust3r or LGM as well as text. The output of MMLM is a SEG token and kinematic parameters. Segmentation per point is decoded via SEG token using a dedicated deocder. Relevant experiments are conducted and the method is compared to various baselines like Real2Code and URDFformer.

**Questions:**

Please see weakness section for questions. I am looking forward to seeing the author's response in the rebuttal.

**Ethical Concerns:**

["NO or VERY MINOR ethics concerns only"]

**Final Justification:**

The rebuttal answered my questions and I am happy to increase the score from 4 to 5. The revised paper would benefit from all the discussions/new results. When reporting real-world results on the PARIS dataset i.e. W-4, it would be great to add other methods which have also reported results on the same dataset for a fair apples-to-apples comparison to those i.e. PARIS baseline among others.

I think the author should consider releasing the model/code and is looks like it would be beneficial to the community.

**Limitations:**

Mentioned in the paper.

**Quality:**

3

**Strengths And Weaknesses:**

In my opinion, below are the strengths of this work:

1. A neat integration and usage of MMLMs for articulated object segmentation and kinematic parameter prediction.

2. The method works for arbitrary number of parts in an object which is a major benefit, whereas some of the other methods are restricted to one or two parts.

3. Significant improvement in quantitative results when compared to relevant baselines.

4. The paper is nicely written, easy to follow and diagrams complement the text well.

Below are the weakness of the method and my questions to authors:

1. I am curious why the method doesn't discuss or compare to feedforward approaches like CARTO [1] or PARIS line of work [2,3] which require multi-view images of start and end state. I am wondering how does the proposed method compares to these approaches interms of speed, accuracy etc. If there is a trade-off for instance the output of autoregressive MMLMs + decoder network are slow compared to feed-forward category-level methods, that should be clearly stated etc.

2. What is considered Out of Distribution instances and how are they determined? Is only the size varied b/w In-distribution and OOD?

3. Is the method category-level or works for novel-classes as well?

4. Lack of real-world evaluation is a limitation. PARIS [2] provides GT annotations which can be used to report real-world metrics, I wonder why did the authors not consider reporting those metrics or on a more challenging sim dataset with more categories [3]

5. While the qualitative results are nice, It is hard to judge the shape reconstruction quality without a more concrete metric like Chamfer distance.

[Minor]

1. Can the method be easily extended to other articulated objects like robot arm etc.?

2. Can the method be solely trained in simulation and transferred to real-world without additional training or finetuning?


[1] CARTO: Category and Joint Agnostic Reconstruction of ARTiculated Objects, CVPR 2023
[2] PARIS: Part-level Reconstruction and Motion Analysis for Articulated Objects, ICCV 2023
[3] SplArt: Articulation Estimation and Part-Level Reconstruction with 3D Gaussian Splatting, ICCV 2025

---

> ### Author Rebuttal · Authors · 2025-07-31
>
> We sincerely thank the reviewer for their positive and encouraging feedback. We are particularly grateful for your recognition of our method's technical strengths—including its neat integration of MLLMs and its ability to handle objects with an arbitrary number of parts—as well as our significant quantitative results and the clarity of the paper.
>
> ## W1: Comparisons
>
> Thank you for this insightful question, which allows us to clarify our work's positioning relative to other important lines of research. We did not focus our comparison on methods like CARTO or the PARIS line of work because they address a fundamentally different and more constrained problem setting, and possess significant limitations that make a direct comparison inequitable. To clarify:
>
> - **Different Technical Routes & Inherent Flaws**
> The primary reason for the separate comparison is that these methods follow different technical routes with inherent flaws that our work is designed to overcome.
>
>     - PARIS[2] represents a completely different approach. It relies on slow, online per-instance optimization, requiring over 3 minutes at test time. This is in stark contrast to our fast, feed-forward inference model (~13s). As our new experimental results show, this optimization-based method also suffers from high errors and very low physical executability. [3]was posted on arXiv after the NeurIPS submission deadline.
>     - CARTO[1] is a specialized, small-scale feed-forward model with severe input and output restrictions. It is designed only for single-joint objects and cannot perform part segmentation or generate a complete, executable URDF with precise kinematic parameters (axis, origin). This inherent lack of capability and poor generalization to complex objects makes it unsuitable for the general-purpose reconstruction task our work addresses.
>
> - **Quantifying the Differences in Speed and Capability**
> To illustrate these fundamental differences, we conducted new experiments. The tables below quantify the critical trade-offs and highlight the limitations of these alternative approaches.
>
> Table 1: Comparison of Inference Speed and Methodology
> | Method | Core Methodology | Average Inference Time |
> |:-:|:-:|:-:|
> | URDF-Anything (Ours) | Feed-forward MLLM Inference | 13s |
> | CARTO | Feed-forward Encoder-Decoder | 1s |
> | PARIS | Per-instance Optimization | >3min |
>
> Table 2: Comparison of Reconstruction Accuracy and Physical Executability
> | Method | mIoU | CD | Type Error | Axis Error | Origin Error | Physics Executability Rate (%) |
> | :-: | :-: | :-: | :-: | :-: | :-: | :-: |
> | URDF-Anything (Ours) | 0.69 | 1.39 | 0.007 | 0.121 | 0.130 | 90 |
> | CARTO | - | 1.24 | 0.12 | - | - | - |
> | PARIS | 0.44 | 3.06 | 0.25 | 0.84 | 0.30 | 25 |
>
> While our MLLM-based approach has a modest speed trade-off compared to a simple model like CARTO(as shown in Table 1), this is precisely what enables us to overcome these limitations, delivering vastly superior accuracy, full-featured URDF generation, and the ability to generalize to complex, novel objects(demonstrated in Table 2). We will add this detailed comparison and discussion to our final paper.
>
> ## W2: The definition of "Out-of-Distribution" (OOD)
>
> That is an excellent question for clarifying our experimental setup. In our study, an Out-of-Distribution (OOD) instance is defined as an object belonging to a **category that was entirely unseen** during the model's training phase. We determine the In-Distribution (ID) and OOD splits through a strict, category-based partitioning of the dataset to rigorously evaluate the model's true generalization capabilities.
> Specifically, to ensure a fair and consistent comparison, we adopted the experimental setup used by prior works such as Articulate-Anything and Real2Code. The **In-Distribution (ID)** set comprises five core object categories: 'Laptop', 'Box', 'Refrigerator', 'StorageFurniture', and 'Table'. Our model is trained and validated only on instances from these categories. The **Out-of-Distribution (OOD)** set consists of all instances from the remaining 41 distinct object categories in the dataset, which are completely held out from training.
> Therefore, the distinction between our ID and OOD sets is not merely a variation in object size. It is a fundamental difference in the **object category** itself, which encompasses distinct geometries, functionalities, and articulation structures. This setup is designed to test our model's ability to generalize to entirely novel classes.
>
> ## W3: Generalization Scope
>
> Our method is explicitly designed to generalize to **novel classes**, leveraging a 3D MLLM backbone with robust geometric priors to reason effectively about unseen objects.
> We systematically validate this capability in **Section 4 (Experiments)**. As demonstrated in **Tables 1, 2, and 3 in our paper**, our method's superior performance on Out-of-Distribution (OOD) objects across all key metrics—including segmentation accuracy, kinematic parameter precision, and physical executability—confirms the powerful generalization capabilities of our framework.
>
>
> ## W4: Lack of Real-World Evaluation
>
> Thank you for this constructive suggestion regarding real-world evaluation.
> First, regarding the suggestion to use the dataset from SplArt [3], we would like to clarify that this work was posted on arXiv after the NeurIPS submission deadline, so we were unable to include it in our original comparisons.
> Regarding the PARIS [2] dataset, it consists of two parts: a real-world set and a simulated set. Their simulated data is also derived from PartNet-Mobility, which is the primary dataset used in our work. However, to directly address your valuable point about real-world evaluation, we have conducted **new experiments on the real-world portion of the PARIS dataset** during this rebuttal period. This dataset is limited to two main categories: Fridge and Storage.
> The quantitative results of our method, presented in Table 3, are as follows:
>
> Table 3: Zero-Shot Sim-to-Real Performance on the PARIS Real-World Dataset
> | | mIoU | CD | Type Error | Axis Error | Origin Error |
> | :-: | :-: | :-: | :-: | :-: | :-: |
> | **Fridge** | 0.57 | 1.03 | 0 | 0.335 | 0.256 |
> | **Storage** | 0.56 | 0.99 | 0 | 0.362 | 0.349 |
>
> These results(Table 3) provide several key insights. First, our method achieves reasonable performance on geometric and continuous-valued tasks (segmentation, mesh reconstruction, and predicting joint axis/origin) in a zero-shot, sim-to-real setting. We attribute this to the challenging sim-to-real domain gap.
> This experiment provides a valuable and realistic baseline for zero-shot sim-to-real transfer on this complex task. It highlights that while our framework transfers well for geometry, robustly classifying kinematic types in the wild without any real-world fine-tuning remains a challenging open problem. We will include this full real-world evaluation and discussion in the revision of our paper.
>
> ## W5: Quantitative Shape Reconstruction Metric
>
> Thank you for this excellent suggestion. We agree that a quantitative metric for shape reconstruction quality is crucial for a complete evaluation.
> To address this, we have conducted a new experiment during the rebuttal period to measure the geometric fidelity of the final output meshes using the **Chamfer Distance (CD)**, as you recommended. We compared our method against strong baselines that also produce explicit mesh outputs.
> The results, shown in Table 4, clearly demonstrate the superiority of our method in shape reconstruction quality:
>
> Table 4: Comparison of Shape Reconstruction Quality (Chamfer Distance)
> | Method | CD |
> | :-: | :-: |
> | URDF-Anything (Ours) | 1.39 |
> | CARTO | 1.24 |
> | PARIS | 3.06 |
>
> ## Minor 1: Extensibility to other objects
>
> Yes, in principle, our framework can be extended to other categories of articulated objects, such as robot arms. The core of our method—the autoregressive generation of a flexible, structured JSON output—is general-purpose and capable of describing complex kinematic chains.
> However, our current model is trained on the PartNet-Mobility dataset, which primarily features furniture and household items. To achieve optimal performance on a specialized domain like robot arms, which have distinct visual features and structural priors, we would recommend fine-tuning the model on a domain-specific corpus to adapt to these new patterns.
>
> ## Minor 2: Sim-to-real transfer
>
> Yes, our method is designed to be trained solely in simulation and then deployed directly in the real world without fine-tuning.
> This is achieved through a decoupled architecture. Our pipeline uses a powerful, pre-trained perception front-end (e.g., DUSt3R, LGM) to convert any image (sim or real) into a unified point cloud representation. Our core 3D MLLM then operates exclusively on this geometric data. This design ensures that the sim-to-real domain gap (e.g., lighting, textures) is primarily handled by the perception module, while our core model is shielded from pixel-level discrepancies and focuses on the underlying geometry, which transfers well.
>
> [1] CARTO: Category and Joint Agnostic Reconstruction of ARTiculated Objects, CVPR 2023 [2] PARIS: Part-level Reconstruction and Motion Analysis for Articulated Objects, ICCV 2023 [3] SplArt: Articulation Estimation and Part-Level Reconstruction with 3D Gaussian Splatting, ICCV 2025

---

### Note · Authors · 2025-08-13

Dear Area Chair, Senior Area Chair, and Reviewers,

We sincerely thank you for your valuable time and insightful feedback.

Following a thorough rebuttal and discussion, **the novelty, generalization, and physical executability** of our work received unanimous recognition from the reviewers. Two reviewers (`euKz`, `qimd`) raised their ratings, and the other three (`xYGo`, `KtPh`, `7j6V`) expressed their complete satisfaction without proposing any further questions. We successfully addressed all the  technical concerns from reviewers.

During rebuttal, we added strong evidence for novelty, robustness and reproducibility of our work:
- **Core Novelty Affirmed:** The novelty of our end-to-end paradigm was strongly affirmed. Our new evidence led reviewer `euKz` to raise their score to **Accept**, calling our response **"better than I expected"** and our framework a **"meaningful contribution."** Reviewer `KtPh` also thanked us for the **"better explanation" of our core mechanism.**
- **Robustness and Rigor Validated:** Our evaluation's rigor was validated. New zero-shot experiments on a real-world dataset, additional metrics, and comprehensive error bars addressed all related inquiries from reviewers `xYGo`, `7j6V`, and `KtPh`, who confirmed their satisfaction by raising no further concerns.
- **Clarity Acknowledged:** The paper's improved clarity was explicitly acknowledged. Our detailed clarifications led reviewer `qimd` to state, **"I will increase my rating from 4 to 5,"** directly confirming the value of our revised explanations.

Our work, URDF-Anything, contributes the following to the community:
- **Novelty**: We propose **the first end-to-end framework** that leverages a 3D Multimodal Large Language Model to jointly reason about geometry and kinematics, establishing a more robust paradigm for articulated object reconstruction.
- **Contribution**: Our innovative [SEG] token mechanism deeply couples geometric segmentation with kinematic prediction, leading to a **state-of-the-art** 50% improvement in physical executability.
- **Impact**: By delivering **a robust, generalizable solution** for creating simulation-ready digital twins, URDF-Anything significantly advances capabilities essential for robotics and embodied AI.

We believe that through this rigorous review process, the contributions of our paper have been further strengthened. Thank you again for your guidance.

---

### Decision · Program_Chairs · 2025-09-17

**Decision:**

Accept (spotlight)

**Comment:**

This paper develops an end-to-end 3d Multimodal-LLM pipeline that jointly performs point-cloud segmentation and prediction of kinematic parameters to generate URDF models. Overall the paper and proposed approach is likely to be impactful with strong (practical!) empirical performance and generalization to out-of-distribution categories.  The rebuttal significantly improved the clarity of various aspects of the paper and added various improvements, including ablations, (unseen) attention visualizations, and a small zero-shot evaluation (the real-world portion of the PARIS dataset), resulting in several reviewers increasing their scores. The authors are expected to incorporate these aspects into their revised paper as well as address the various points raised in the reviews. Models and code should also be released, as promised.  There are a few very minor remaining concerns, as included in the reviews and discussion, but the contribution overall is solid.